**Nocturnal Atmospheric Synergistic Oxidation Reduces the Formation of Low-volatility Organic Compounds from Biogenic Emissions**

Han Zang[1], Zekun Luo[1], Chenxi Li[1], Ziyue Li[1], Dandan Huang[2,*], Yue Zhao[1,*]

[1]School of Environmental Science and Engineering, Shanghai Jiao Tong University, Shanghai, 200240, China

[2]Shanghai Academy of Environmental Sciences, Shanghai, 200233, China

*Correspondence: Yue Zhao (yuezhao20@sjtu.edu.cn); Dandan Huang (huangdd@saes.sh.cn);

**Abstract**
Volatile organic compounds (VOCs) are often subject to synergistic oxidation by different oxidants
in the atmosphere. However, the exact synergistic oxidation mechanism of atmospheric VOCs and
its role in particle formation remain poorly understood. In particular, the reaction kinetics of the key
reactive intermediates, organic peroxy radicals ($RO_2$), during synergistic oxidation is rarely studied.
Here, we conducted a combined experimental and kinetic modelling study of the nocturnal
synergistic oxidation of α-pinene (the most abundant monoterpene) by $O_3$ and $NO_3$ radicals as well
as its influences on the formation of highly oxygenated organic molecules (HOMs) and particles.
We find that in the synergistic $O_3$ + $NO_3$ regime, where OH radicals are abundantly formed via
decomposition of ozonolysis-derived Criegee intermediates, the production of $C_xH_yO_z$-HOMs is
substantially suppressed compared to that in the $O_3$-only regime, mainly because of the depletion of
of α-pinene $RO_2$ derived from ozonolysis and OH oxidation by those arising from $NO_3$ oxidation
via cross reactions. Measurement-model comparisons further reveal that the cross-reaction rate
constants of $NO_3$-derived $RO_2$ with $O_3$-derived $RO_2$ are on average 10 – 100 times larger than those
of $NO_3$-derived $RO_2$ with OH-derived $RO_2$. Despite a strong production of organic nitrates in the
synergistic oxidation regime, the substantial decrease of $C_xH_yO_z$-HOM formation leads to a
significant reduction in ultralow- and extremely low-volatility organic compounds, which
significantly inhibits the formation of new particles. This work provides valuable mechanistic and
quantitative insights into the nocturnal synergistic oxidation chemistry of biogenic emissions and
will help to better understand the formation of low-volatility organic compounds and particles in
the atmosphere.

## 1. Introduction

The Earth's atmosphere is a complex oxidizing environment in which multiple oxidants coexist. During the nighttime, $NO_3$ radicals (generated by the reaction of $NO_2$ and $O_3$) and $O_3$ contribute significantly to the oxidation of volatile organic compounds (VOCs) (Huang et al., 2019), while during the daytime, the fast photolysis of $NO_3$ radicals and rapid photochemical formation of OH radicals and $O_3$ make the latter two the major oxidants for VOCs (Zhang et al., 2018). Therefore, the degradation of ambient VOCs is subject to concurrent oxidation by different oxidants. Gas-phase oxidation of VOCs from biogenic emissions (BVOCs) by these major atmospheric oxidants produces a key type of reactive intermediates, organic peroxy radicals ($RO_2$), a portion of which can undergo fast autoxidation forming a class of highly oxygenated organic molecules (HOMs) with low volatilities (Jokinen et al., 2014; Mentel et al., 2015; Berndt et al., 2016; Zhao et al., 2018; Iyer et al., 2021; Shen et al., 2022; Ehn et al., 2014). HOMs typically contain six or more oxygen atoms, and play a key role in the formation of atmospheric new particles and secondary organic aerosol (SOA) (Kirkby et al., 2016; Berndt et al., 2018; Zhao et al., 2018; Ehn et al., 2014; Bianchi et al., 2019), which have important influences on air quality (Huang et al., 2014), public health (Pye et al., 2021), and Earth's radiative forcing (Shrivastava et al., 2017).

Due to the complexity of oxidation mechanisms of BVOCs, previous laboratory studies typically featured only one oxidant and a single SOA precursor (Berndt et al., 2016; Berndt, 2021; Claflin et al., 2018; Iyer et al., 2021; Boyd et al., 2015). However, the synergistic oxidation by different oxidants may significantly alter the fate of $RO_2$ intermediates, therefore influencing the formation of HOMs and SOA (Bates et al., 2022). Recently, a field study at a boreal forest site in Finland observed a series of nitrate-containing HOM-dimers from the coupled $O_3$ and $NO_3$ oxidation of monoterpenes (Zhang et al., 2020). At the same site, Lee et al. (2020) found that the synergistic oxidation of BVOCs by OH radicals and $O_3$ contributed to the largest fraction of SOA. These studies suggest that the synergistic oxidation of BVOCs by different oxidants plays an important role in the formation of HOMs and SOA in the atmosphere and highlight the needs to investigate the synergistic oxidation mechanisms of BVOCs for a better representation of atmospheric particle formation.

Several laboratory studies have attempted to address the role of synergistic oxidation of BVOCs in the formation of new particles and SOA (Kenseth et al., 2018; Inomata, 2021; Liu et al., 2022; Li et

al., 2024). Kenseth et al. (2018) identified a suite of dimer esters in flow tube experiments that can
be only formed from the OH and $O_3$ synergistic oxidation of β-pinene. These dimers exhibit
extremely low volatility and contributed 5.9 – 25.4% to the total β-pinene SOA. Similarly, Inomata
(2021) found that the presence of OH radicals during α-pinene ozonolysis is a key factor for the
production of low-volatility organic species and significantly promotes new particle formation
(NPF). On the other hand, the addition of $O_3$ in the monoterpene photooxidation system also
significantly increases the SOA mass yield (Liu et al., 2022). In addition, a recent chamber study
by Bates et al. (2022) showed that the synergistic oxidation of α-pinene by $NO_3$ radicals and $O_3$ can
significantly enhance the SOA yield compared to the $NO_3$ + α-pinene regime, which has nearly 0%
SOA yield (Fry et al., 2014; Hallquist et al., 1999; Mutzel et al., 2021), and they revealed that the
SOA yield in the $NO_3$ + $O_3$ oxidation system largely depends on the $RO_2$ fates. Most recently, Li et
al. (2024) found that during α-pinene ozonolysis, the presence of nitrooxy-$RO_2$ radicals formed from
$NO_3$ oxidation can significantly suppress the production of ultralow-volatility organic compounds
(ULVOCs) and thereby NPF. These laboratory studies together provide growing evidence that
synergistic oxidation of BVOCs by different oxidants have profound impacts on atmospheric
particle formation. However, the specific synergistic mechanisms of different oxidants and
oxidation pathways remain obscure. Although a few studies underscored the importance of the $RO_2$
fates (Bates et al., 2022; Li et al., 2024), the exact interactions between $RO_2$ species derived from
different oxidants are still unclear, and quantitative constraints on the reaction rate of different $RO_2$
species are quite limited.
Here we conducted an investigation of the synergistic $O_3$ + $NO_3$ oxidation of α-pinene, one of the
most abundant monoterpenes in the atmosphere, using a combination of laboratory experiments and
detailed kinetic modelling, and focusing on the fate of $RO_2$ intermediates arising from different
oxidation pathways. The α-pinene oxidation experiments were conducted in a custom-built flow
reactor. The molecular composition of $RO_2$ species and HOMs in different oxidation regimes was
characterized using a chemical ionization atmospheric pressure interface time-of-flight mass
spectrometer (CI-APi-ToF) employing a nitrate ion source. The measured distributions of specific
$RO_2$ and HOMs across different oxidation regimes were fitted with a kinetic model using Master
Chemical Mechanisms (MCM v3.3.1) updated with recent advances of α-pinene $RO_2$ chemistry
(Wang et al., 2021; Iyer et al., 2021; Shen et al., 2022; Zang et al., 2023), which allows for
quantitative constraints on $RO_2$ kinetics and synergistic oxidation mechanisms. Atmospheric
relevance of the experimental results was evaluated by modelling the investigated oxidation
chemistry under typical nocturnal atmospheric conditions.
**2. Materials and Methods**
**2.1 Flow tube experiments**
Experiments of α-pinene oxidation in different regimes (i.e., synergistic $O_3 + NO_3$ oxidation vs. $O_3$-
only) were carried out under room temperature (298 K) and dry (relative humidity < 5%) conditions
in a custom-built flow tube reactor (FTR, Figure S1). $O_3$ and $NO_2$ were added into a glass tube
(Figure S1) to form $NO_3$ radical and its precursor $N_2O_5$:
$NO_2 + O_3 \rightarrow NO_3 + O_2$ (R1)
$NO_3 + NO_2 \leftrightarrow N_2O_5$ (R2)
$O_3$ was generated by passing a flow of ultra-high-purity (UHP) $O_2$ (Shanghai Maytor Special Gas
Co., Ltd.) through a quartz tube housing a pen-ray mercury lamp (UV-S2, UVP Inc.) and its
concentration was measured by an ozone analyzer (T400, API). $NO_2$ was obtained from a gas
cylinder (15.6 ppm, Shanghai Weichuang Standard Gas Co., Ltd.). The initial $NO_2$ concentration in
the flow tube was 4.5 – 6.4 ppb. To prevent the titration of $NO_3$ radicals by NO, all the experiments
were performed without the addition of NO. The total air flow in the $NO_3$ generation glass tube was
0.6 L min$^{-1}$ and 0.4 L min$^{-1}$ for the gas-phase HOM and SOA formation experiments, respectively.
The produced $N_2O_5$ and $NO_3$ radicals, as well as the excessive $O_3$ were added into the FTR to initiate
α-pinene oxidation. For the $O_3$-only experiments, only $O_3$ was added into FTR.
The α-pinene gas was generated by evaporating a defined volume of its liquid (99%, Sigma-Aldrich)
into a cleaned and evacuated canister (SilcoCan, RESTEK), and then added into FTR through a
movable injector at a flow rate of 22 – 108 mL min$^{-1}$. The initial concentration of α-pinene in the
flow reactor ranged from 100 – 500 ppb. In some experiments, the gas of cyclohexane (~ 100 ppm),
which was generated by bubbling a gentle flow of UHP $N_2$ through its liquid (LC-MS grade, CNW),
was added into the flow reactor as a scavenger of OH radicals formed from α-pinene ozonolysis.
For experiments characterizing the formation of HOMs, the total air flow in the FTR was 10.8 L
min$^{-1}$ and the residence time was 25 seconds. The short reaction time and the small amount of
reacted α-pinene (see Table S1) in these experiments prevented the formation of particles. For the
experiments characterizing the formation of SOA particles, a larger FTR was used, with a total air
flow of 5 L min$^{-1}$ and a residence time of 180 seconds. A summary of the conditions including the
simulated concentrations of $NO_2$, $N_2O_5$ and $NO_3$ radicals, as well as the concentration of α-pinene
oxidized by each oxidant in different experiments are shown in Table S1.
The gas-phase $RO_2$ radicals and closed-shell products were measured using a nitrate-based CI-APi-
ToF (abbreviated as nitrate-CIMS; Aerodyne Research, Inc.), which has been described in detail
previously (Zang et al., 2023). A long ToF-MS with a mass resolution of ∼10000 Th/Th was used
here. The mass spectra within the m/z range of 50 – 700 were analyzed using the tofTools package
developed by Junninen et al. (2010) based on Matlab. The total ion counts (TIC) with values of (5.9
– 6.2) $\times 10^4$ cps are similar under different reaction conditions. In this study, we assume that the
$C_xH_yO_z$-HOMs derived from ozonolysis and OH oxidation of α-pinene exhibit the same sensitivity
in nitrate-CIMS. However, the highly oxygenated organic nitrates (ONs) may have a significantly
lower sensitivity compared to the $C_xH_yO_z$-HOM counterparts, given that the substitution of -OOH
or -OH groups by –$ONO_2$ group in the molecule would reduce the number of H-bond donors, which
is a key factor determining the sensitivity of nitrate-CIMS (Shen et al., 2022; Hyttinen et al., 2015).
Recently, Li et al. (2024) used CI-Orbitrap with ammonium or nitrate reagent ions to detect
oxygenated organic molecules in the synergistic $O_3$ + $NO_3$ regime and found that both the ion
intensity of ONs and their signal contribution to the total dimers were much lower when using nitrate
as reagent ions.
A scanning mobility particle sizer (SMPS, TSI), consisting of an electrostatic classifier (model
3082), a condensation particle counter (model 3756), and a long or nano differential mobility
analyzer (model 3081 and 3085) with a measurable size range of 4.6 – 156.8 nm or 14.6 – 661.2
nm, respectively, was employed to monitor the formation of particles in the flow tube. During the
HOM formation experiments, even under conditions with the highest initial α-pinene concentration
(500 ppb), only a tiny amount of particles was formed, with mass concentrations of (6.4 $\pm$1.7) $\times 10^{-3}$
$^{3}$ and (1.0 $\pm$0.4) $\times 10^{-2}$ μg m$^{-3}$ and number concentrations of 574 $\pm$148 and 256 $\pm$68 cm$^{-3}$ in the $O_3$-
only (Exp 5) and $O_3$ + $NO_3$ regimes (Exp 11), respectively. These results suggest that the formation
of SOA particles in the HOM formation experiments is negligible and would have no significant
influence on the fate of $RO_2$ and closed-shell products.
**2.2 Estimation of HOM volatility**
A modified composition-activity method was used to estimate the saturation mass concentration
($C^*$) of HOMs in this study according to the approach developed by Li et al. (2016):
$$\log_{10}C^* = (n_C^0 - n_C)b_C - n_O b_O - 2\frac{n_C n_O}{n_C + n_O}b_{co} - n_N b_N - n_S b_S$$
where $n_C^0$ is the reference carbon number; $n_C$, $n_O$, $n_N$, and $n_S$ are the atom numbers of carbon,
oxygen, nitrogen, and sulfur, respectively; $b_C$, $b_O$, $b_N$, and $b_S$ are the contribution of each atom
to $\log_{10}C^*$, respectively; $b_{co}$ is the carbon–oxygen nonideality (Donahue et al., 2011). These $b$-
values were provided by Li et al. (2016).
It should be noted that the CHON compounds used in the data set by Li et al. (2016) are mostly
amines, amides, and amino acids, and only contain a limited number of ONs (0.07%). Since different
types of CHON compounds have very different vapor pressures (Isaacman-Vanwertz and Aumont,
2021), this formula-based approach can be biased to estimate the $C^*$ of ONs. Considering that the
$–ONO_2$ and –OH groups have similar impacts on vapor pressure and that the CHON species are
predominantly ONs in our study, all $–ONO_2$ groups are treated as –OH groups during the estimation
of vapor pressure (Daumit et al., 2013; Isaacman-Vanwertz and Aumont, 2021).
Gas-phase HOMs are grouped into five classes based on their $\log_{10}C^*$ (Donahue et al., 2012;
Bianchi et al., 2019; Schervish and Donahue, 2020), that is, ULVOCs ($\log_{10}C^* < -8.5$), extremely
low-volatility organic compounds (ELVOCs, $-8.5 < \log_{10}C^* < -4.5$), low-volatility organic
compounds (LVOCs, $-4.5 < \log_{10}C^* < -0.5$), semi-volatile organic compounds (SVOCs, $-0.5 <$
$\log_{10}C^* < 2.5$), and intermediate-volatility organic compounds (IVOCs, $2.5 < \log_{10}C^* < 6.5$).
**2.3 Kinetic model simulations**
Model simulations of specific $RO_2$ radicals and closed-shell HOMs formed in different oxidation
regimes were performed to constrain the reaction kinetics and mechanisms using the Framework
for 0-D Atmospheric Modeling (F0AM v4.1) (Wolfe et al., 2016), which employs MCM v3.3.1
(Jenkin et al., 2015). The α-pinene oxidation mechanism was updated with the state-of-the-art
knowledge on the chemistry of $RO_2$ autoxidation and cross reactions forming HOM monomers and
dimers, respectively (Zhao et al., 2018; Wang et al., 2021; Iyer et al., 2021; Shen et al., 2022). The
detailed updates have been described in our previous study (Zang et al., 2023). In particular, the
formation and subsequent reactions of the ring-opened primary $C_{10}H_{15}O_4$-$RO_2$, the highly
oxygenated acyl $RO_2$, as well as the $C_{10}H_{15}O_2$-$RO_2$ arising from H-abstraction by OH radicals
during α-pinene ozonolysis are included in the model according to recent studies (Iyer et al., 2021;
Zhao et al., 2022; Zang et al., 2023; Shen et al., 2022).
To investigate the synergistic reactions of $RO_2$ derived from the oxidation of α-pinene by different
oxidants, we added the cross reactions of the primary nitrooxy-$RO_2$ derived from $NO_3$ oxidation
($^{NO3}RO_2$), i.e., $C_{10}H_{16}NO_5$-$RO_2$, with $RO_2$ derived from ozonolysis ($^{CI}RO_2$) and OH oxidation
($^{OH}RO_2$). Recently, Zhao et al. (2018) revealed the bulk rate constant for $^{CI}RO_2$ and $^{OH}RO_2$ self/cross
reactions to be $2 \times 10^{-12}$ $cm^3$ molecule$^{-1}$ s$^{-1}$, and Bates et al. (2022) constrained the rate constant for
$^{NO3}RO_2$ self/cross reactions to be $1 \times 10^{-13} - 1 \times 10^{-12}$ $cm^3$ molecule$^{-1}$ s$^{-1}$. In the present study, the
default rate constant for $^{NO3}RO_2 + {}^{CI}RO_2$ was set to $2 \times 10^{-12}$ $cm^3$ molecule$^{-1}$ s$^{-1}$, the same to that for
self/cross reactions of $^{CI}RO_2$ and $^{OH}RO_2$. The default rate constant for $^{NO3}RO_2 + {}^{NO3}RO_2$ was set to
$1 \times 10^{-12}$ $cm^3$ molecule$^{-1}$ s$^{-1}$. The ratio of the cross-reaction rate constant of $^{NO3}RO_2 + {}^{CI}RO_2$ to that
of $^{NO3}RO_2 + {}^{OH}RO_2$ was tuned to achieve a good measurement-model agreement for the distribution
of specific $RO_2$ and HOMs across different oxidation regimes. Recent studies suggested that the
ROOR' dimer formation rates from the highly oxygenated $RO_2$ are fast (Berndt et al., 2018; Molteni
et al., 2019). As a result, a relatively high dimer formation branching ratio of 50% was used for
different $RO_2$ (e.g., $^{CI}RO_2$, $^{OH}RO_2$, $^{NO3}RO_2$) in the model, except for the reaction of $^{NO3}RO_2 + {}^{NO3}RO_2$,
for which ROOR' dimer formation was not considered, given the extremely low signals of $CHON_2$
dimers observed in the synergistic oxidation regime (see Section 3.1). With these default kinetic
parameters, the $RO_2$ bimolecular lifetimes were predicted to be $10.9 - 25.9$ s in the $O_3$-only regime
and $8.4 - 11.8$ s in the $O_3 + NO_3$ regime in the HOM formation experiments. Considering that the
$RO_2$ cross-reaction kinetics remain highly uncertain, sensitivity analyses were performed to evaluate
their influences on the results in this study (see Section 3.2). Previous studies indicated that the
primary $^{NO3}RO_2$ radicals arising from α-pinene are prone to lose the nitrate group and form
pinonaldehyde with high volatility (Kurtén et al., 2017; Fry et al., 2014). Therefore, we did not
consider the autoxidation of primary $^{NO3}RO_2$ in the model. Considering the presence of $NO_2$ in the
experiments, the reactions of $RO_2 + NO_2 \leftrightarrows ROONO_2$ were also included in the model (Zang et al.,

205     2023).

**3. Results and Discussion**
**3.1 Molecular distribution of $RO_2$ and HOMs in the synergistic oxidation regime**
The abundance of gas-phase $RO_2$ species and HOMs in different oxidation regimes is shown in
Figure 1a. The species signals are normalized by the total reacted α-pinene in each regime.
Compared to the $O_3$-only regime, the normalized signals of total $RO_2$ and HOMs decrease by 63 –
68% in the synergistic $O_3 + NO_3$ regime. Although $NO_3$ oxidation accounts for a considerable
fraction of reacted α-pinene in the synergetic oxidation regime, the signal contributions of HOM-
ONs are not significant. This might be due to the low sensitivity of nitrate-CIMS to the ONs formed
involving $NO_3$ oxidation (Section 2.1). Although there remain considerable uncertainties in
instrument sensitivities to different compounds, sensitivity analyses suggest that varying the CIMS
sensitivities to $RO_2$ and HOMs by a factor of 10 would not significantly influence their relative
distribution across different oxidation regimes (see Section S1 for details).
Note that the initial concentrations of α-pinene and $O_3$ in the two oxidation regimes were the same.
In addition, model simulations show that in the synergistic $O_3 + NO_3$ regime, over 97% of OH
radicals react with α-pinene and the depletion of OH by $NO_2$ is minor (0.2 – 1.3%). Also, $NO_3$
radicals almost entirely (over 98.5%) react with α-pinene and their reaction with $RO_2$ has negligible
influence on the fate of $RO_2$ (Figure S2). Meanwhile, the depletion of acyl $RO_2$ by $NO_2$ only leads
to a small reduction (4 – 5% and 7 – 12%, respectively) in total $C_xH_yO_z$-HOM monomers and dimers
in the synergistic regime compared to the $O_3$-only regime. As a result, the strong reduction in HOM
formation due to the presence of $NO_3$ oxidation is likely mainly due to (i) the fast competitive
consumption of α-pinene by $NO_3$ radicals, which leads to a reduction in the reacted α-pinene by $O_3$
($\Delta[\alpha\text{-pinene}]_{O_3}$, Figure S3) and thereby $C_xH_yO_z$-HOM signals, and (ii) the cross reactions of $^{CI}RO_2$
or $^{OH}RO_2$ with $^{NO3}RO_2$, which suppress the autoxidation and self/cross reactions of $^{CI}RO_2$ and $^{OH}RO_2$
to form $C_xH_yO_z$-HOMs.
To quantify the contribution of cross reactions of $^{NO3}RO_2$ with $^{CI}RO_2/^{OH}RO_2$ to the suppressed
formation of $C_xH_yO_z$-HOMs in the synergistic oxidation regime, $C_xH_yO_z$-HOM signals shown in
Figure 1a are first normalized to $\Delta[\alpha\text{-pinene}]_{O3}$ in each oxidation regime and then compared between
different oxidation regimes (see Figure 1b). Notably, after excluding the influence of reduced $\Delta[\alpha\text{-}$
$\text{pinene}]_{O3}$, the $C_xH_yO_z$-HOM signals still drop by $32 - 33\%$ in the $O_3 + NO_3$ regime compared to
those in the $O_3$-only regime, indicating a significant contribution of the coupled reactions between
$^{NO3}RO_2$ and $^{Cl}RO_2$ or $^{OH}RO_2$ to suppressed $C_xH_yO_z$-HOM formation.

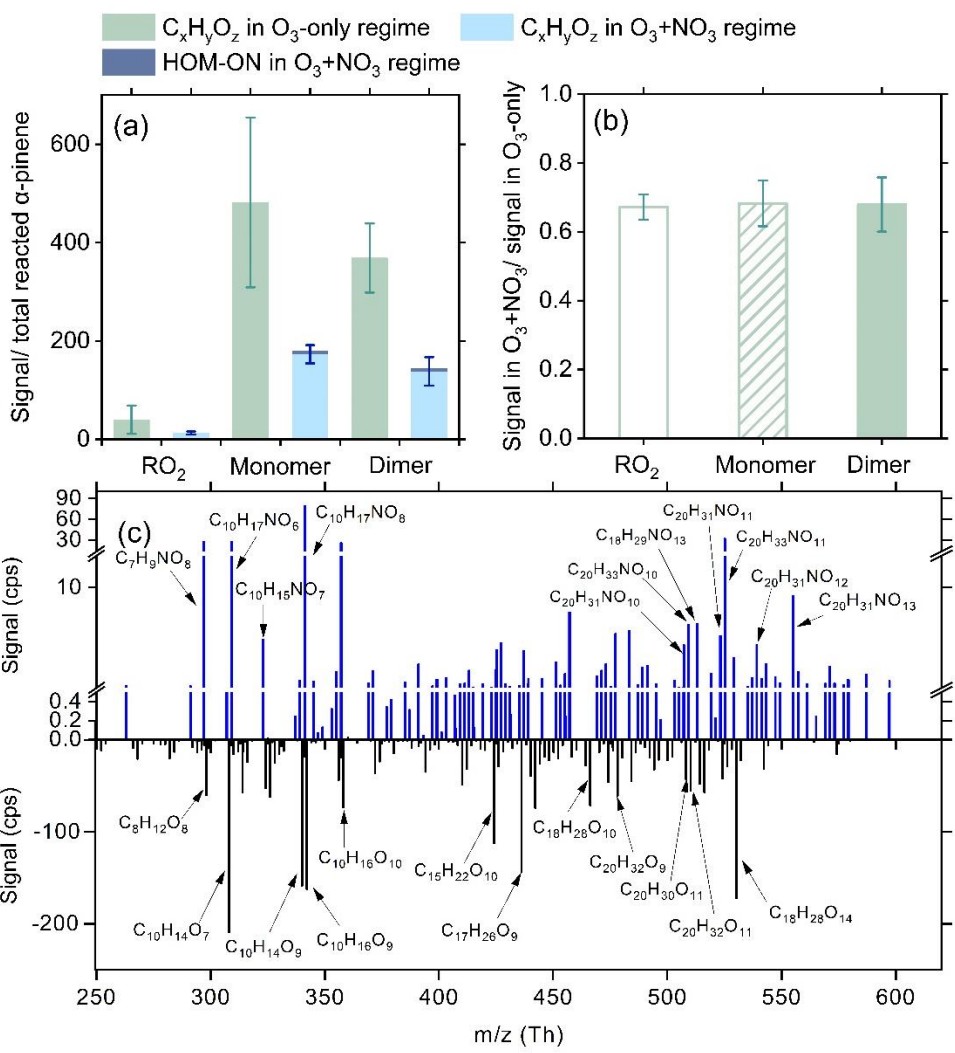

Figure 1 Distributions of $RO_2$ and HOMs in the $O_3$-only and $O_3 + NO_3$ regimes. (a) Signals of total
$RO_2$, as well as HOM monomers and dimers normalized by the total reacted $\alpha$-pinene in each
oxidation regime (Exps 1-5, 7-11). (b) Relative changes in the normalized signals of $C_xH_yO_z$-HOMs
in the $O_3 + NO_3$ regime versus the $O_3$-only regime. Ion signals are normalized to $\Delta[\alpha\text{-pinene}]_{O3}$ in
each oxidation regime to highlight the suppression effect of the synergistic chemistry between
$^{NO3}RO_2$ and $^{Cl}RO_2$ or $^{OH}RO_2$ on $C_xH_yO_z$-HOM formation. (c) Difference mass spectrum between
the two oxidation regimes. The positive and negative peaks indicate the species with enhanced and
decreased formation in the $O_3 + NO_3$ regime compared to the $O_3$-only regime, respectively.
Figure 1c shows a difference mass spectrum highlighting the changes in species distribution
between the two oxidation regimes. Almost all $C_xH_yO_z$-HOM species decrease significantly in the
$O_3 + NO_3$ regime compared to the $O_3$-only regime. Besides, a large set of HOM-ON species are
formed, despite their relatively low signals. It should be noted that no obvious signals of highly
oxygenated $^{NO3}RO_2$ ($C_{10}H_{16}NO_x$, $x \geq 6$) were observed by nitrate-CIMS in the $O_3 + NO_3$ oxidation
system. One possible reason is that nitrate-CIMS exhibits relatively low sensitivity to the ONs.
Secondly, the instrument's mass resolution is not high enough to differentiate the mass closure
between some of $^{NO3}RO_2$ and $C_xH_yO_z$-HOMs with strong peaks (Table S2), limiting the detection
of $^{NO3}RO_2$ species. Furthermore, previous studies revealed that the primary $^{NO3}RO_2$ radicals (i.e.,
$C_{10}H_{16}NO_5$-$RO_2$) in the α-pinene + $NO_3$ system mainly react to form pinonaldehyde (Kurtén et al.,
2017; Perraud et al., 2010). It is likely that only a very small amount of $^{NO3}RO_2$ can undergo
intramolecular H-shift/$O_2$ addition to form highly oxygenated $^{NO3}RO_2$. It should be pointed out that
although the primary $C_{10}H_{16}NO_5$-$RO_2$ species arising from $NO_3$ oxidation may not undergo fast
autoxidation, they tend to efficiently terminate $^{Cl}RO_2$ and/or $^{OH}RO_2$ and suppress the formation of
$C_xH_yO_z$-HOMs.
As shown in Figure 1c, although several closed-shell monomeric HOM-ONs have been observed in
the synergistic oxidation regime, only a few of them exhibit relatively high signals. Among them,
$C_{10}H_{17}NO_8$ may be formed by the autoxidation of $C_{10}H_{16}NO_6$-$RO_2$ derived from the intramolecular
H-shift of primary $^{NO3}RO$ radicals ($C_{10}H_{16}NO_4$-RO). In addition, although CI is a soft ionization
method, the fragmentation of chemically labile species still occurs during the ionization in nitrate-
CIMS. It is possible that some of dimeric HOM-ONs are fragmented to $C_{10}H_{17}NO_8$ during nitrate-
CIMS measurements. In a recent study by Li et al. (2024), $C_{10}H_{17}NO_8$ was also identified during
the synergistic oxidation of α-pinene by $O_3$ and $NO_3$. However, the exact origin of this species
remains to be clarified.
The $C_{20}$ dimers with only one nitrogen atom are very likely to be formed from the cross reactions
of $^{Cl}RO_2$ or $^{OH}RO_2$ with $^{NO3}RO_2$, which provides direct evidence for the synergistic $RO_2$ chemistry
in the $O_3 + NO_3$ regime. The $CHON_2$ dimers were also observed in the $O_3 + NO_3$ regime, despite
their much lower signals than CHON dimers, which is different from the recent studies by Bates et
al. (2022) and Li et al. (2024), which found $CHON_2$ dimers account for an important fraction of the
total dimer signals in the synergistic oxidation regime. A potential explanation for this discrepancy
is the difference in the instrument sensitivity in these studies (Section 2.1). In general, the nitrate-
CIMS has lower sensitivities to ONs than to the $C_xH_yO_z$-HOM counterparts (Shen et al., 2022;
Hyttinen et al., 2015). Bates et al. (2022) used $CF_3O^-$ as the reagent ion of CIMS. Its sensitivity to
ONs might be significantly higher than the nitrate ion. In addition, Li et al. (2024) observed a
significantly lower signal contribution of $CHON_2$ dimers using CI-Orbitrap with nitrate reagent ions
than with ammonium ions. Despite both using nitrate reagent ions, the nitrate CI-Orbitrap in Li et
al. (2024) possibly exhibits higher sensitivities to ONs than the nitrate-CIMS in our study.
**3.2 Synergistic reaction efficiencies of different $RO_2$ species**
In the $O_3 + NO_3$ regime, synergistic reactions are likely to occur between $^{Cl}RO_2$, $^{OH}RO_2$ and $^{NO3}RO_2$.
Figure 2 shows the $\Delta[\alpha\text{-pinene}]_{O3}$-normalized signal ratios of specific $C_{10}$ $RO_2$ as well as their
related $C_xH_yO_z$-HOM monomers and dimers in the synergistic $O_3 + NO_3$ regime vs. the $O_3$-only
regime. It should be noted that the second-generation oxidation processes are strongly inhibited by
the excess of $\alpha$-pinene in this study, thus the predominant type of $RO_2$ observed here is primary $RO_2$.
Model simulations show that the H-abstraction of $\alpha$-pinene by OH radicals contributes less than 2%
to the formation of $C_{10}H_{15}O_x$-$RO_2$ and related HOMs under different experimental conditions
(Figure S5). Therefore, $C_{10}H_{15}O_x$-$RO_2$ observed in this study are primarily $^{Cl}RO_2$. Notably, the
$^{Cl}RO_2$ ($C_{10}H_{15}O_x$) and related $C_{10}H_{14}O_x$-HOMs decrease by ~20 – 80% in the $O_3 + NO_3$ regime
(Figures 2 a, b), while the decreasing extent of $^{OH}RO_2$ ($C_{10}H_{17}O_x$) and related $C_{10}H_{18}O_x$-HOMs are
significantly smaller (0 – 30%). In particular, some of the most oxygenated $C_{10}H_{17}O_x$-$RO_2$ and
$C_{10}H_{18}O_x$-HOMs ($x \geq 9$) even increase unexpectedly in the synergistic oxidation regime. For the
$C_{10}H_{16}O_x$-HOMs that can be derived from the self/cross reactions of both $^{Cl}RO_2$ and $^{OH}RO_2$, their
reductions are at a medium level. Because of the very small contribution of acyl $RO_2$ to the total $C_{10}$
$RO_2$ (0.4%) (Zang et al., 2023), their consumption by $NO_2$ leads to less than 2% reduction in the
$C_{10}$ $^{Cl}RO_2$ signals. Therefore, the more significant decrease in signals of $^{Cl}RO_2$ and related HOMs
as compared to the OH-derived ones in the synergistic $O_3 + NO_3$ regime is primarily due to the more
efficient cross reactions of $^{NO3}RO_2$ with $^{Cl}RO_2$ than with $^{OH}RO_2$. Because a large amount of $^{Cl}RO_2$
is terminated by $^{NO3}RO_2$, fewer $^{Cl}RO_2$ are available to terminate $^{OH}RO_2$. As a result, more $^{OH}RO_2$
can undergo autoxidation to form highly oxygenated $C_{10}H_{17}O_x$-$RO_2$ and $C_{10}H_{18}O_x$-HOMs ($x \geq 9$),
leading to an increase in signals of these species. Consistently, the signals of $C_{20}$ HOM dimers
decrease by ~20 – 40% in the $O_3$ + $NO_3$ regime compared to that in $O_3$-only regime, and the signal
reduction of dimers ($C_{20}H_{30}O_x$) formed by $^{Cl}RO_2$ is slightly larger than that of the dimers ($C_{20}H_{34}O_x$)
arising from $^{OH}RO_2$ (Figure 2c). Note that the highly oxygenated $C_{20}H_{34}O_x$ dimers ($x \geq 13$) that can
be formed from self/cross reactions of $C_{10}H_{17}O_x$-$RO_2$ ($x \geq 9$) are not observed in this study, likely
due to their low abundance and the limitation of instrument sensitivity.

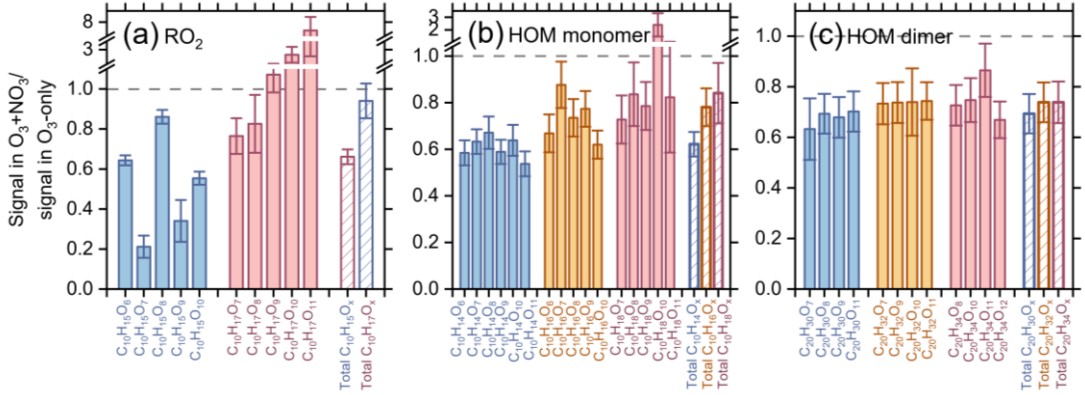


Figure 2. Normalized signal ratios of (a) specific and total $C_{10}H_{15,17}O_x$-$RO_2$ radicals, as well as their
related (b) $C_{10}$ HOM monomers and (c) $C_{20}$ HOM dimers in the $O_3$ + $NO_3$ regime vs. the $O_3$-only
regime (Exps 1-5, 7-11). Ion signals observed in each oxidation regime are normalized to $\Delta[\alpha$-
pinene$]_{O3}$.
The above results are somewhat different from the most recent study by Li et al. (2024), which
found that the measured $C_{10}H_{15}O_x$-$RO_2$ increased slightly with $NO_3$ radicals while $C_{10}H_{17}O_{5,7}$-$RO_2$
from OH chemistry decreased by a factor of 9. Li et al. (2024) indicated that additional $C_{10}H_{15}O_x$
could be produced from the H-abstraction pathway of $NO_3$ oxidation of $\alpha$-pinene. However, in the
monoterpene oxidation system, the rate constant for H-abstraction by $NO_3$ radicals is $(4 - 10) \times 10^{-17}$
$cm^3$ molecule$^{-1}$ s$^{-1}$, which is $10^3 - 10^4$ times lower than that for the $NO_3$ addition channel (Martinez
et al., 1998). Besides, the subsequent reactions of $RO_2$ species formed from H-abstraction by $NO_3$
radicals should be very similar to those derived from H-abstraction by OH radicals, which was found
not important for $C_xH_yO_z$-HOM formation in the absence of NO (Zang et al., 2023). Therefore, the
H-abstraction of $\alpha$-pinene by $NO_3$ radicals would have negligible influence on $C_{10}H_{15}O_x$ formation.
As Li et al. (2024) used a low $\alpha$-pinene concentration and relatively high $O_3$ and $NO_3$ concentrations
in their experiments, the secondary oxidation of aldehydes, such as the substantially formed
pinonaldehyde, by $NO_3$ radicals might be important, which could contribute to the additional
formation of $C_{10}H_{15}O_x$-$RO_2$. However, as noted above, the second-generation oxidation processes
are strongly inhibited due to the excess of α-pinene in this study, therefore the formation of
secondary $C_{10}H_{15}O_x$-$RO_2$ is not important.
In addition, Li et al. (2024) reported that the fraction of α-pinene oxidized by OH radicals decreased
from 44% in the $O_3$ oxidation system to 6% in the $O_3 + NO_3$ system, mainly due to the depletion of
OH radicals by $NO_2$ and the competitive consumption of α-pinene by $NO_3$ radicals, which resulted
in a significant decrease in $C_{10}H_{17}O_{5,7}$ radicals from OH chemistry as observed in their experiments.
However, in the present study, because of the excess of α-pinene, over 97% of OH radicals react
with α-pinene and the depletion of OH by $NO_2$ is minor (0.2 – 1.3%) in the $O_3 + NO_3$ regime. The
reduction in the reacted α-pinene by OH radicals is less than 10% compared to the $O_3$-only regime.
As a result, a smaller decrease in $C_{10}H_{17}O_{5,7}$ radicals was observed in our study.
To gain quantitative constraints on the relative reaction efficiency of $^{NO3}RO_2 + ^{Cl}RO_2$ vs. $^{NO3}RO_2 +$
$^{OH}RO_2$ (i.e., $k_{NO3+Cl}/k_{NO3+OH}$), the signal ratios of $C_{10}$-$^{Cl}RO_2$ and $^{OH}RO_2$ as well as their related $C_{10}$
HOMs in the synergistic oxidation regime vs. the $O_3$-only regime were predicted using a kinetic
model (see Section 2.3) with different $k_{NO3+Cl}/k_{NO3+OH}$ ratios. Figure 3 shows a measurement-model
comparison of those signal ratios. When the ratio of $k_{NO3+Cl}/k_{NO3+OH}$ is smaller than or equal to 1,
the simulated signal ratios of many $RO_2$ and HOMs differ significantly from the measured ratios,
especially for some $C_{10}H_{17}O_x$-$RO_2$ and $C_{10}H_{18}O_x$-HOMs. When the ratio of $k_{NO3+Cl}/ k_{NO3+OH}$ is 10 –
100, there is a good measurement-model agreement for most of $RO_2$ and HOMs. Therefore, we
conclude that the cross-reaction rate constants of $^{NO3}RO_2 + ^{Cl}RO_2$ are on average 10 – 100 times
larger than those for $^{NO3}RO_2 + ^{OH}RO_2$. This different $RO_2$ cross-reaction efficiency is the main
reason for the significantly larger decrease in the abundance of $^{Cl}RO_2$ and related HOMs as
compared to the OH-derived ones (see Figure 2).

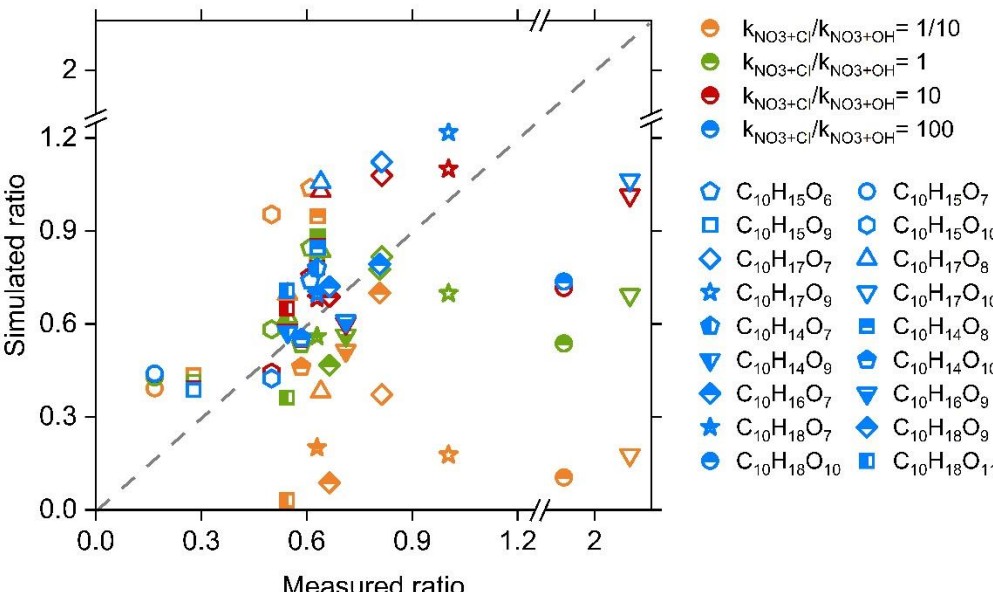

Figure 3. Measurement-model comparisons of the signal ratios of different $C_{10}$ $RO_2$ and HOMs in the synergistic $O_3$ + $NO_3$ regime vs. the $O_3$-only regime. The cross-reaction rate constant of $^{NO3}RO_2$ + $^{Cl}RO_2$ was set to $2 \times 10^{-12}$ $cm^3$ molecule$^{-1}$ s$^{-1}$ and the rate of $^{NO3}RO_2$ + $^{OH}RO_2$ was varied from $2 \times 10^{-11}$ $cm^3$ molecule$^{-1}$ s$^{-1}$ to $2 \times 10^{-14}$ $cm^3$ molecule$^{-1}$ s$^{-1}$ in the model.

As a competitive reaction pathway, the autoxidation rates of $RO_2$ can affect the extent to which $RO_2$ cross reactions influence the $RO_2$ fate and HOM formation. Therefore, sensitivity analyses of the autoxidation rate of $RO_2$ were conducted to evaluate its influence on the changes of $RO_2$ and related HOM concentrations in the synergistic $O_3$ + $NO_3$ regime vs. the $O_3$-only regime (Figure S6). In these analyses, a $k_{NO3+Cl}/k_{NO3+OH}$ ratio of 10 was used according to the above discussions. As the autoxidation rate of $^{OH}RO_2$ increases from 0.28 to 10 s$^{-1}$, corresponding to the rate range reported in previous studies (Berndt et al., 2016; Zhao et al., 2018; Xu et al., 2019), the simulated reduction of highly oxygenated $^{OH}RO_2$ and related $C_{10}H_{18}O_x$-HOMs in the synergistic $O_3$ + $NO_3$ regime exhibits a slight decrease (< 10%) but still agrees reasonably well with the measured value (Figures S6 a-d). Considering that the autoxidation rates of $^{Cl}RO_2$ used in the model approach their upper limits reported in the literature, i.e., ~1 s$^{-1}$ for the butyl ring-opened $C_{10}H_{15}O_4$-$RO_2$ (Iyer et al., 2021) and relatively smaller rates for ring-retained $C_{10}H_{15}O_4$-$RO_2$ (0.02 – 0.29 s$^{-1}$, see Scheme S1) (Zhao et al., 2021), we also lowered the autoxidation rate constants of $^{Cl}RO_2$ by a factor of 10 to see its influence on $RO_2$ and HOM distribution in the $O_3$ + $NO_3$ regime. The simulated reduction of $^{Cl}RO_2$ and $C_{10}H_{14}O_x$-HOMs in this case decreases by 7 – 16% (Figures S6 e-h), while that of $C_{10}H_{16}O_x$-HOMs increases by up to 31% (Figures S6 i, j). However, the simulated results are still close to the

measured values. These sensitivity analyses suggest that the uncertainty in the autoxidation rates of
$^{OH}RO_2$ and $^{Cl}RO_2$ could slightly affect the simulated distribution of $RO_2$ and HOMs across different
oxidation regimes but not significantly change the $k_{NO3+Cl}/k_{NO3+OH}$ ratio obtained in this study.
Further sensitivity analyses on the rate constant and dimer formation branching ratio of $RO_2$ cross
reactions indicate that the uncertainties in these reaction kinetics do not alter the conclusion
regarding the $k_{NO3+Cl}/k_{NO3+OH}$ ratio either (see details in Sections S2 and S3).
Cyclohexane was added in some experiments as an OH scavenger to elucidate the role of $^{OH}RO_2$
chemistry in HOM formation in the $O_3 + NO_3$ regime. In the presence of cyclohexane, most of
$^{OH}RO_2$ ($C_{10}H_{17}O_x$) and related HOM monomers ($C_{10}H_{18}O_x$) and dimers ($C_{20}H_{32}O_x$ and $C_{20}H_{34}O_x$)
decrease by more than 70% (Figure 4), while $^{Cl}RO_2$ ($C_{10}H_{15}O_x$) and related HOM monomers
($C_{10}H_{14}O_x$) only decrease slightly. Accordingly, the reduction in $C_{20}H_{32}O_x$ and $C_{20}H_{34}O_x$ dimers is
significantly larger than that of $C_{20}H_{30}O_x$. These results are in a good agreement with previous
measurements (Zhao et al., 2018; Zang et al., 2023). The $C_{10}H_{16}O_x$ species, which can arise from
both $^{Cl}RO_2$ and $^{OH}RO_2$, exhibit a medium reduction (Figure 4b). It is interesting to note that with the
addition of cyclohexane, there is a significant increase in $C_{20}H_{31}NO_x$, which are formed from the
cross reactions of $^{Cl}RO_2$ with $^{NO3}RO_2$. Such an enhanced production of $C_{20}H_{31}NO_x$ as compared to
the slightly deceased formation of $C_{20}H_{30}O_x$ indicates that the $^{Cl}RO_2 + {}^{NO3}RO_2$ reactions are
competitive compared to the $^{Cl}RO_2 + {}^{Cl}RO_2$ and $^{Cl}RO_2 + {}^{OH}RO_2$ reactions. As a result, when the
$^{OH}RO_2$ are depleted, the $^{Cl}RO_2$ that are supposed to react with $^{OH}RO_2$, efficiently react with $^{NO3}RO_2$
to form $C_{20}H_{31}NO_x$, leading to the increase in $C_{20}H_{31}NO_x$ signals. Consistent with the experimental
measurements, the model simulations show that the concentrations of $C_{20}H_{31}NO_x$ in the $O_3 + NO_3$
regime increase with the addition of cyclohexane as an OH scavenger (Figure S9). However, the
simulated enhancement is slightly lower than the measurements, which might be due to the
uncertainties in the $RO_2$ cross-reaction kinetics in the model.

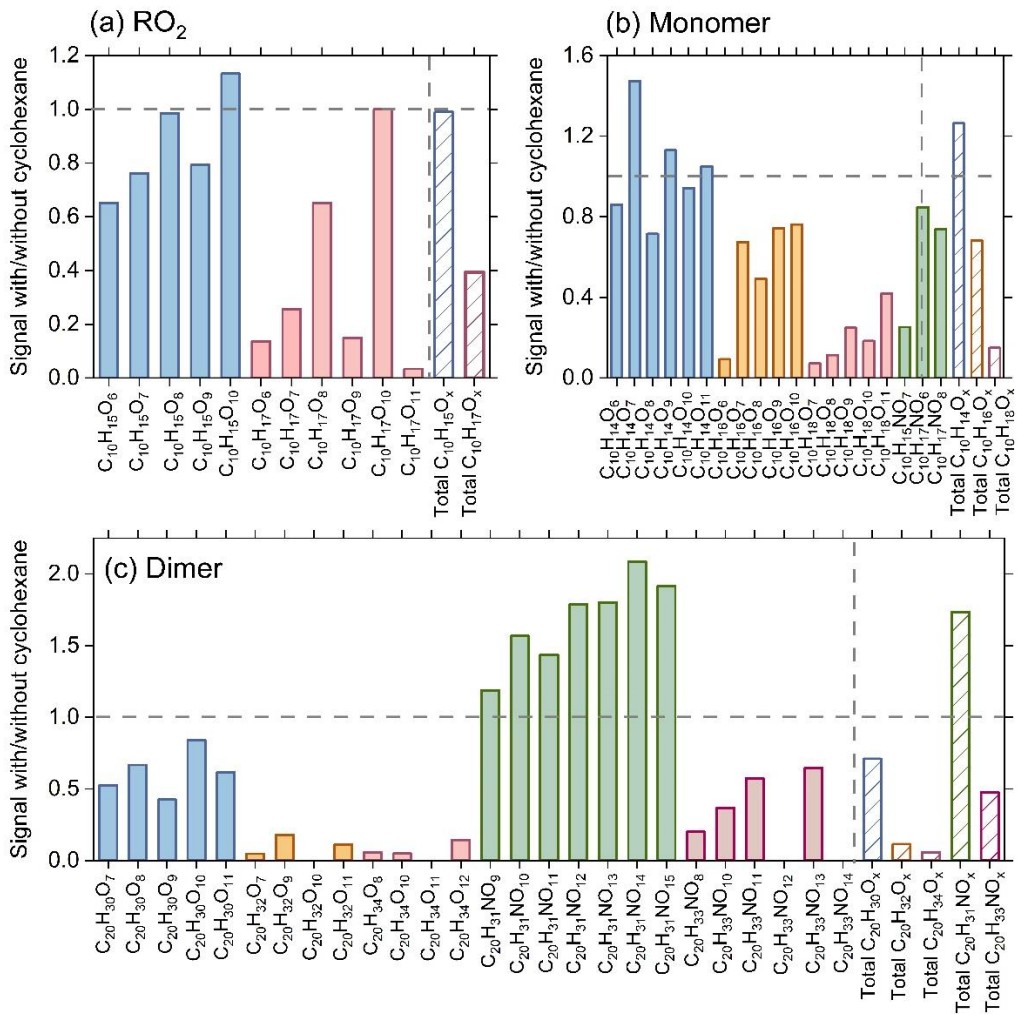

Figure 4. Relative changes in signals of (a) $C_{10}$ RO$_2$, (b) $C_{10}$ HOMs, and (c) $C_{20}$ dimers due to the addition of 100 ppm cyclohexane as an OH scavenger derived in the synergistic $O_3$ + NO$_3$ regime (Exps 6 and 12).

**3.3 Influence of synergistic oxidation on low-volatility organics and particle formation**

Compared to the $O_3$-only regime, there are a remarkable reduction in $C_xH_yO_z$-HOMs and a strong formation of HOM-ONs due to the efficient cross reactions between $^{NO3}RO_2$ and $^{Cl}RO_2$ in the synergistic oxidation regime. This significant change in HOM composition and abundance would alter the volatility distribution of HOMs and influence the formation of particles. The volatilities of HOMs formed in the two oxidation regimes are estimated using a modified composition-activity method (see Section 2.2) and shown in Figure 5. The abundance of $C_xH_yO_z$-HOMs characterized as ULVOCs and ELVOCs decreases considerably in the synergistic $O_3$ + NO$_3$ regime compared to the $O_3$-only regime (Figure 5), in agreement with the very recent observations by Li et al. (2024) who found that the presence of NO$_3$ radicals during α-pinene ozonolysis significantly reduced the

abundance of ULVOCs. Although substantial amounts of HOM-ONs are formed in the $O_3 + NO_3$
regime, they generally have higher volatilities (i.e., characterized as ELVOCs to IVOCs) (Figure 5).
Therefore, the synergistic $O_3 + NO_3$ oxidation of α-pinene significantly reduces the formation of
ULVOCs and increases the overall volatility of total HOMs.

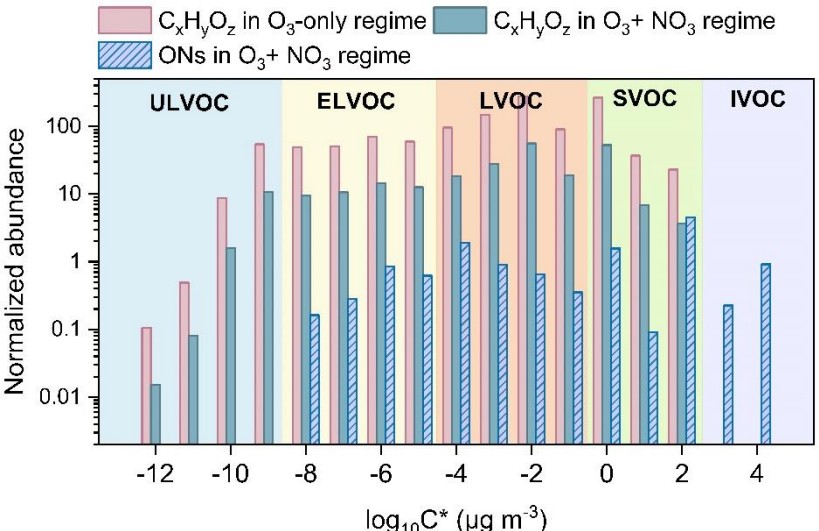


Figure 5. Volatility distribution of $C_xH_yO_z$-HOMs and HOM-ONs formed in the $O_3 + NO_3$ regime
and $O_3$-only regime (Exps 1, 7). Ion signals in each oxidation regime are normalized to the
corresponding total reacted α-pinene.
Figure 6a shows the particle number and mass concentrations formed in the two oxidation regimes
in SOA formation experiments (Exps 13, 14). The particle number concentration decreases by ~50%
whereas the particle mass concentration increases by a factor of 2 in the synergistic $O_3 + NO_3$ regime,
compared to that in the $O_3$-only regime. The presence of $NO_3$ radicals during α-pinene ozonolysis
reduces the abundance of ULVOCs, which are the key species driving particle nucleation, thereby
leading to a reduction in the particle number concentration in the $O_3 + NO_3$ regime. On the other
hand, substantial formation of HOM-ONs is expected from the cross reactions of $^{NO3}RO_2$ with $^{Cl}RO_2$
and $^{OH}RO_2$ in the synergistic oxidation regime (Li et al., 2024; Bates et al., 2022), although their
signals are relatively low due to the low sensitivity of nitrate-CIMS to ONs in this study. The newly
formed HOM-ONs have relatively higher volatilities and are inefficient in initiating particle
nucleation, but they are able to partition into the formed particles and contribute to the particle mass
growth. Meanwhile, as the particle number concentration decreases drastically in the synergistic
oxidation regime, more condensable vapors are available for each particle to grow to larger sizes
(Figure 6b), which would in turn favor the condensation of more volatile organic species including
ONs due to the reduced curvature effect of the larger particles, ultimately resulting in an increase in
SOA mass concentrations.

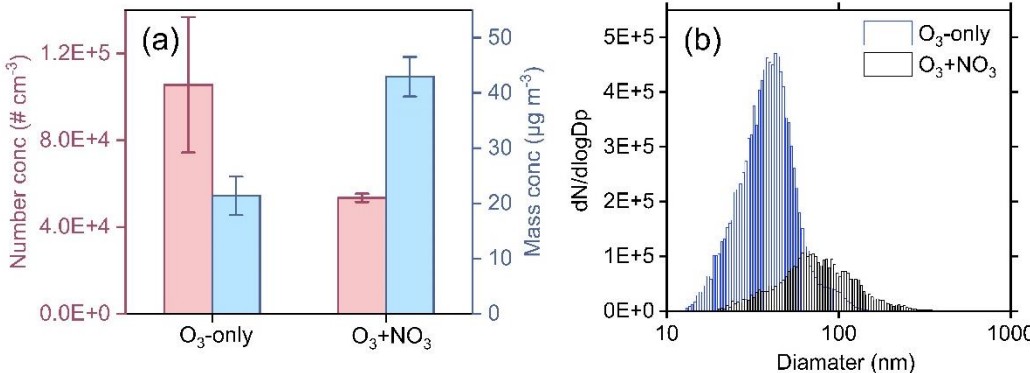


Figure 6. Number and mass concentrations (a), as well as the size distribution (b) of particles formed
from the ozonolysis and synergistic $O_3$ + $NO_3$ oxidation of α-pinene (Exps 13-14).
Recently, Bates et al. (2022) also found that in chamber experiments with seed particles, the SOA
mass yields were significantly higher during α-pinene oxidation by $O_3$ + $NO_3$ than during ozonolysis,
mainly due to the substantial formation and condensation of dimeric ONs. However, in the absence
of seed particles, synergistic $O_3$ + $NO_3$ oxidation of α-pinene does not nucleate in their study. This
phenomenon might be due to the high concentrations of $NO_2$ (72 ppb) and $O_3$ (102 ppb) as well as
the relatively low concentration of α-pinene (27 ppb) in their experiments. As indicated by Bates et
al. (2022), under this conditions $NO_3$ radicals were substantially formed and contributed to a
dominant fraction (75%) of α-pinene oxidation, which strongly inhibited the production of low-
volatility species and particle nucleation.
**3.4 Atmospheric relevance of experimental results**
In the present study, the flow tube experiments were conducted under dry conditions. Although
water vapor may affect the fate of Criegee intermediates and $RO_2$ radicals and thereby HOM
formation during the oxidation of organics under humid conditions, there is growing evidence that
such effects in the α-pinene oxidation system are small. Kinetics studies have found that the
stabilized Criegee intermediates (SCIs) arising from α-pinene ozonolysis can undergo fast
unimolecular decay at a rate constant of $60 - 250$ s$^{-1}$ (Vereecken et al., 2017; Newland et al., 2018),
which is rapid compared to their reaction with water vapor, in particular for syn-SCIs, under
atmospheric conditions (Vereecken et al., 2017; Newland et al., 2018). In addition, the yield of OH
radicals from Criegee decomposition is independent of RH (Atkinson et al., 1992; Aschmann et al.,
2002). Consistent with the fast unimolecular reaction kinetics revealed by these studies, recent
laboratory measurements have shown that the contribution of SCIs to the formation of gas-phase
and particle-phase dimers are small (<20%) during α-pinene ozonolysis (Zhao et al., 2018; Zhao et
al., 2022). Furthermore, the molecular composition and abundance of HOM monomers and dimers
(Li et al., 2019) and the formation of particle-phase dimers (Zhang et al., 2015; Kenseth et al., 2018)
do not change significantly with RH ranging from 3% to 92%. These studies suggest that the
humidity condition does not strongly affect the HOM formation chemistry in the α-pinene
ozonolysis system.
To evaluate the relevance of our experimental findings to the real atmosphere, we performed
chemical model simulations of HOM formation from nocturnal synergistic $O_3$ + $NO_3$ oxidation of
α-pinene under typical atmospheric conditions. In these simulations, constant concentrations of α-
pinene (1 ppb), $O_3$ (30 ppb), NO (5 ppt), $NO_2$ (1.8 ppb), $NO_3$ radicals (0.2 or 1 ppt), OH radicals (5
$- 50 \times 10^4$ molecules cm$^{-3}$), $HO_2$ radicals (4 ppt), as well as a constant RH of 50% and temperature
of 298 K were used as typical nocturnal conditions in the boreal forest according to the field studies
(Stone et al., 2012; Lee et al., 2016a; Brown and Stutz, 2012; Geyer et al., 2003b; Kristensen et al.,
2016; Hakola et al., 2012; Liebmann et al., 2018). Considering the rapid deposition of oxidized
biogenic compounds (Nguyen et al., 2015), a typical dilution lifetime of 5 h (i.e., $k_{dil}$ = 1/5 h$^{-1}$) was
assumed in the model. According to the above analysis, the cross-reaction rate constants for $^{NO3}RO_2$
+ $^{CI}RO_2$ and $^{NO3}RO_2$ + $^{OH}RO_2$ were set to 2 $\times 10^{-12}$ cm$^3$ molecule$^{-1}$ s$^{-1}$ and 2 $\times 10^{-13}$ cm$^3$ molecule$^{-1}$
s$^{-1}$ in the model, respectively. The formation of $RO_2$ with oxygen numbers higher than 11 was not
considered in the model, due to the large uncertainty in the autoxidation rate constants of the highly
oxygenated $RO_2$. In fact, the autoxidation rate of the highly oxygenated $RO_2$ is expected to be small
given the significant decrease in the number of active sites for intramolecular H-abstraction in the
molecule. As a result, the contribution of the most oxygenated HOMs to the total HOM monomers
could be relatively small (Zhao et al., 2018; Claflin et al., 2018).
In the absence of $NO_3$ radicals (with $NO_3$ concentrations and formation rates set to zero), the amount
of α-pinene consumed during 4 hours of simulation is 1.04 ppb. When a relatively low $NO_3$
concentration (0.2 ppt) is considered (Figure 7a), the amount of α-pinene consumed is 1.48 ppb, and
the ozonolysis is the primary loss pathway of α-pinene (68%), followed by $NO_3$ (30%) and OH
oxidation (2%). The reactions of $RO_2$ + $HO_2$, $RO_2$ + NO, and $RO_2$ + $RO_2$ account for ~49%, ~27%,
and ~24% of the total $RO_2$ fate, respectively (Figure S10a). Compared to the ozonolysis of α-pinene,
the synergistic $O_3$ + $NO_3$ oxidation leads to a reduction of 3% and 13% in the formation of $C_xH_yO_z$-
HOM monomers and dimers, respectively (Figure 7b). Given that the concentrations of α-pinene
and oxidants were held constant during the simulation, the consumptions of α-pinene by $O_3$ and OH
radicals are the same across different oxidation regimes. Therefore, the decreases in the
concentrations of $C_xH_yO_z$-HOM monomers and dimers in the presence of $NO_3$ oxidation are mainly
due to the cross reactions of $^{NO3}RO_2$ with other $RO_2$. When the $NO_3$ concentration is as high as 1
ppt as reported in field studies (Liebmann et al., 2018), the consumption of α-pinene reaches 3.24
ppb, of which 68% is contributed by $NO_3$ oxidation (Figure 7c). Under this condition, the $RO_2$ +
$RO_2$ reactions account for ~34% of the total $RO_2$ fate (Figure S10b). As a result, the cross reactions
of $^{NO3}RO_2$ with other $RO_2$ play a more important role in the HOM formation. The production of
$C_xH_yO_z$-HOM monomers and dimers decreases by 12% and 43%, respectively, due to the presence
of $NO_3$ oxidation (Figure 7d). We note that the variation in RH from 0 – 90% in the model has
negligible influence on the relative changes in $C_xH_yO_z$-HOMs under these nocturnal atmospheric
conditions (Figure S11). Considering that there are uncertainties in the dilution rate constant, a
sensitivity analysis was performed by varying the $k_{dil}$ in the range of 0.04 – 0.2 $h^{-1}$. It is found that
the variation within these rate values does not significantly influence the response of $C_xH_yO_z$-HOM
dimer formation to concurrent $NO_3$ oxidation (Figure S12).

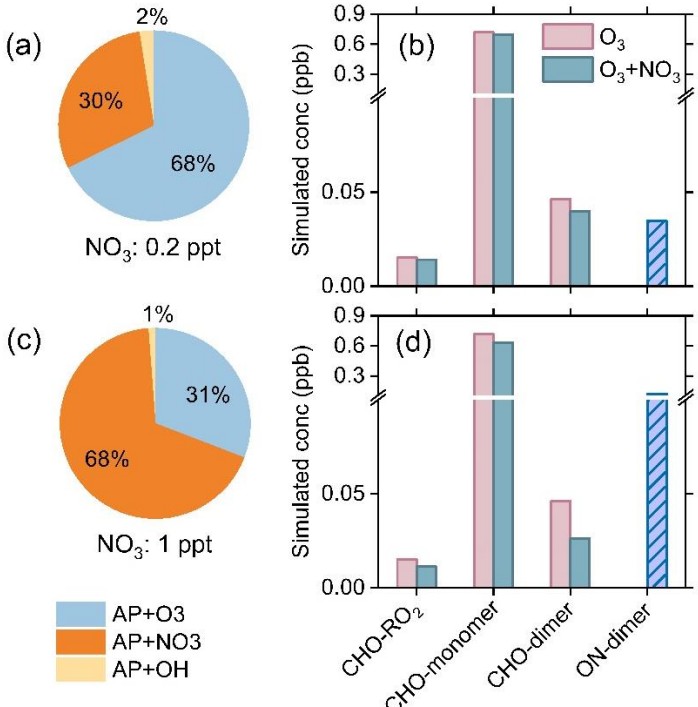

Figure 7. Model simulations of α-pinene oxidation and HOM formation under typical nighttime conditions in the boreal forest. (a, c) Contributions of different loss pathways of α-pinene by different oxidants at $NO_3$ concentrations of 0.2 and 1 ppt, respectively; (b, d) Concentrations of $C_xH_yO_z$-HOMs and HOM-ONs formed by synergistic $O_3$ + $NO_3$ oxidation and ozonolysis of α-pinene under conditions corresponding to (a) and (c). The simulations were run for 4 h after an 8-h spin-up for intermediates and secondary species.

Field observations have shown that $NO_3$ radicals, $O_3$, and OH radicals all had important contributions to monoterpene oxidation during the early morning after sunrise and late afternoon before sunset in the southeastern United States (Zhang et al., 2018). In addition, relatively high nighttime OH concentrations of $(2 - 10) \times 10^5$ molecules $cm^{-3}$ were measured in some areas such as Germany and New York City (Faloona et al., 2001; Geyer et al., 2003a). As a result, a model simulation was conducted using a 10 times higher OH concentration ($5 \times 10^5$ molecules $cm^{-3}$). The concentration of $NO_3$ radicals is 1 ppt and the concentrations of other species are the same as the values mentioned above. With a higher OH concentration, $O_3$, $NO_3$, and OH radicals account for 28%, 61%, and 11% to the total α-pinene consumption, respectively (Figure S13 a). Compared to the results under low OH concentration, the formation of $C_xH_yO_z$-HOM monomers and dimers are all enhanced under high OH concentration (Figure S13 b). This is mainly due to the promoted self/cross reactions of $^{OH}RO_2$, as well as the promoted formation of $C_{10}H_{15}O_x$-RO_2 derived from H-abstraction pathway by OH radicals. Nevertheless, the presence of $NO_3$ oxidation still reduces the

formation of $C_xH_yO_z$-HOM dimers by 26% (Figure S13 b).
Furthermore, model simulations under typical conditions in the southeastern United States (see
details in Section S4) suggest that the coexistence of isoprene appears to exacerbate the suppression
effect of synergistic oxidation on HOM formation from monoterpenes. As shown in Figure S14, in
the absence of isoprene, the synergistic $O_3 + NO_3$ oxidation of α-pinene leads to a reduction of 13%
and 24% in the formation of $C_xH_yO_z$-HOM monomers and dimers, respectively. When isoprene is
present, as the isoprene + $NO_3$ oxidation produces a significant amount of nitrooxy $RO_2$ that can
also scavenge α-pinene-derived $^{Cl}RO_2$ and $^{OH}RO_2$ via cross reactions, the synergistic oxidation leads
to a slightly larger reduction in $C_xH_yO_z$-HOM monomers and dimers (15% and 31%, respectively).
The above model simulations suggest that under nocturnal atmospheric conditions with a very low
$NO_3$ concentration, the $RO_2$ radical pool is dominated by $^{Cl}RO_2$ and their self/cross reactions are a
major contributor to ULVOCs such as the highly oxygenated $C_{20}$ dimers as observed in boreal forest
(Bianchi et al., 2017). When the $NO_3$ concentration is high, the production of $^{NO3}RO_2$ becomes
significant and their cross reactions with $^{Cl}RO_2$ would suppress the formation of ULVOCs. Although
HOM-ON dimers are readily produced by cross reactions between $^{NO3}RO_2$ and $^{Cl}RO_2$, they generally
have higher volatilities than $C_xH_yO_z$-HOM dimers and therefore are less efficient in initiating
particle formation. However, these HOM-ONs can be an important contributor to the particle mass
growth. As suggested by the model simulations in Bates et al. (2022), the $NO_3$ oxidation of α-pinene
led to a particulate nitrate yield of 7% under nocturnal atmospheric conditions in rural Alabama
during the SOAS campaign. Our results offer mechanistic and quantitative insights on how the
synergistic oxidation of α-pinene by $O_3$ and $NO_3$ radicals can influence the formation of low-
volatility organic compounds and hence particle formation and growth. They also provide a potential
explanation for field observations that NPF events frequently occur in monoterpene-rich regions
during daytime but not at nighttime (Mohr et al., 2017; Kulmala et al., 2001; Junninen et al., 2017).
**4. Conclusions**
This study provides a comprehensive characterization of the nocturnal synergistic oxidation of α-
pinene by $O_3$ and $NO_3$ radicals and its influence on the formation of HOMs and low-volatility
organic compounds using a combination of flow reactor experiments and detailed kinetic model
simulations. It is found that the formation of $C_xH_yO_z$-HOMs in the $O_3 + NO_3$ regime is significantly

suppressed compared to that in the $O_3$-only regime, mainly due to the depletion of ozonolysis-derived $RO_2$ (i.e., $^{Cl}RO_2$ and $^{OH}RO_2$) by $^{NO3}RO_2$ via cross reactions. In addition, the decreases in the abundance of $^{Cl}RO_2$ and related HOMs are significantly larger than those of OH-derived ones, indicating that the $^{NO3}RO_2$ species react more efficiently with $^{Cl}RO_2$ than with $^{OH}RO_2$. Detailed measurement-model comparisons for the distribution of a suite of $^{Cl}RO_2$, $^{OH}RO_2$, and associated HOMs across different oxidation regimes further reveal that the cross reactions between $^{Cl}RO_2$ and $^{NO3}RO_2$ are averagely $10 - 100$ times more efficient than those of $^{OH}RO_2$ and $^{NO3}RO_2$.

The suppressed formation of $C_xH_yO_z$-HOMs in the synergistic $O_3 + NO_3$ regime results in a significant reduction in ULVOCs. Although substantial amounts of HOM-ONs are formed from the cross reactions between $^{NO3}RO_2$ and $^{Cl}RO_2$ or $^{OH}RO_2$ in the synergistic oxidation regime, they have higher volatilities and are less likely to participate in the formation and initial growth of new particles. As a result, in our experiment the formation of new particles in the synergistic oxidation regime is substantially inhibited compared to the $O_3$-only regime. Chemical model simulations further confirm that the synergistic oxidation of α-pinene by $O_3$ and $NO_3$ radicals can significantly inhibit the formation of $C_xH_yO_z$-HOMs, especially the ultra-low volatility $C_xH_yO_z$-HOM dimers under typical nighttime atmospheric conditions. Our study sheds lights on the synergistic oxidation mechanism of biogenic emissions and underscores the importance of considering this chemistry for a better depiction of the formation of low-volatility organics and particles in the atmosphere.

*Data availability.* The data presented in this work are available upon request from the corresponding author.

*Author contributions.* YZ and HZ designed the study, HZ and DH performed the experiments. YZ and HZ analyzed the data, conducted model simulations, and wrote the paper. All other authors contributed to discussion and writing.

*Competing interests.* The authors declare no conflict of interest.

*Acknowledgments.* This work was supported by the National Natural Science Foundation of China (grants 22376137 and 22022607). Dan Dan Huang acknowledges the financial support from the Science and Technology Commission of Shanghai Municipality (grant 21230711000).

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
