# Peer review of "Nocturnal Atmospheric Synergistic Oxidation Reduces the Formation of Low-volatility"

_EGUsphere, 2024_

## Author Comment (AC1)

We are grateful to the reviewer for the thoughtful comments on the manuscript. Our point-to-point responses to each comment are as follows (reviewer's comments are in black font and our responses are in blue font).

General Comments

This work studied nocturnal oxidation of alpha-pinene synergistically by $O_3$, $NO_3$, and OH. The manuscript reports that in the synergistic $O_3 + NO_3$ regime, CHO-HOM production is substantially suppressed compared to $O_3$-only regime, due to rapid termination reactions between $RO_2$ formed from alpha-pinene $+ NO_3$ and those formed from ozonolysis and OH oxidation, which is 10-100 times faster. This effect also leads to a reduction in ultralow and extremely low-volatility organic compounds. The work is solid and well written. However, there are a few issues and unclear details that need to be addressed before published at ACP.

Specific Comments

1. Line 23 in Abstract. Stating that termination reactions are "10-100 times more efficient" is vague. In the kinetic model later, does it assume that the difference is only about $RO_2 + RO_2$ reaction rate constant, but not about dimer yields from these reactions?

Response: Thanks for the reviewer's comment. In the kinetic model, the difference is only about the cross-reaction rate constant of $RO_2 + RO_2$ and not the dimer formation yields.

We have modified the description as "Measurement-model comparisons further reveal that the cross-reaction rate constants of $NO_3$-derived $RO_2$ with $O_3$-derived $RO_2$ are on average 10 – 100 times larger than those of $NO_3$-derived $RO_2$ with OH-derived $RO_2$."

2. Line 117. A reaction time of 25 seconds is long enough to form particles in precursors' concentrations are high. Was particle measurement performed for this?

Response: We used a scanning mobility particle sizer (SMPS, TSI) employing both long and nano differential mobility analyzers (model 3081 and 3085), with a measurable size range of 4.61 – 156.8 nm and 14.6 – 661.2 nm, respectively, to monitor particle formation in the flow tube. Even under conditions with the highest initial α-pinene concentration (500 ppb), only a tiny amount of particles was formed, with mass concentrations of $(6.4 \pm 1.6) \times 10^{-3}$ and $(1.0 \pm 0.3) \times 10^{-2}$ μg m$^{-3}$ and number concentrations of $574 \pm 138$ and $256 \pm 68$ cm$^{-3}$ in the $O_3$-only regime (Exp 5) and $O_3 + NO_3$ regime (Exp 11), respectively. These results suggest that the formation of SOA particles in the HOM formation experiments is negligible and would have no significant influence on the fate of $RO_2$ and closed-shell products.

We have added the results of the particle size measurements to Section 2.1 of the revised manuscript.

"A scanning mobility particle sizer (SMPS, TSI), consisting of an electrostatic classifier (model 3082), a condensation particle counter (model 3756), and a long or nano differential mobility analyzer (model 3081 and 3085) with a measurable size range of 4.61 – 156.8 nm and 14.6 – 661.2 nm respectively, was employed to monitor the formation of particles in the flow tube. During the HOM formation experiments, even under conditions with the highest initial α-pinene concentration (500 ppb), only a tiny amount of particles was formed, with mass concentrations of $(6.4 \pm 1.6) \times 10^{-3}$ and $(1.0 \pm 0.3) \times 10^{-2}$ μg m$^{-3}$ and number concentrations of $574 \pm 138$ and $256 \pm 68$ cm$^{-3}$ in the $O_3$-only regime (Exp 5) and $O_3 + NO_3$

regime (Exp 11), respectively. These results suggest that the formation of SOA particles in the HOM formation experiments is negligible and would have no significant influence on the fate of $RO_2$ and closed-shell products."

3. Section 2.1. A few important details should be provided in this section: (1) under the mixed $O_3/NO_3$ condition, how much of alpha-pinene was oxidized by either oxidant? (2) Was $NO_2$ also present when alpha-pinene was oxidized? (3) What was the typically reacted alpha-pinene concentrations? (4) A model-based estimation of $RO_2$ bimolecular lifetime under these conditions should be provided. And (5) Did the authors assume that in $NO_3$-CIMS, all HOM species have the same sensitivity?

Response: Thanks for the reviewer's comments.

(1) We have added the concentration of α-pinene oxidized by each oxidant in Table S1 in the Supplement.

(2) $NO_2$ was present in the experiments, and we have added a description of its concentration in Section 2.1.

"The initial $NO_2$ concentration in the flow tube was ~4.5 ppb."

(3) The total reacted α-pinene under different experimental conditions is also provided in Table S1.

(4) We have added the model-predicted $RO_2$ bimolecular lifetimes under different experimental conditions in Section 2.3.

"With these default kinetic parameters, the $RO_2$ bimolecular lifetimes were predicted to be 10.9 – 25.9 s in the $O_3$-only regime and 8.4 – 11.8 s in the $O_3$ + $NO_3$ regime in the HOM formation experiments."

(5) In this study, we assume that the $C_xH_yO_z$-HOMs derived from ozonolysis and OH oxidation of α-pinene exhibit the same sensitivity in nitrate-CIMS. However, the highly oxygenated organic nitrates may have a significantly lower sensitivity compared to the $C_xH_yO_z$-HOM counterparts, given that the substitution of -OOH or -OH groups by –$ONO_2$ group in the molecule would reduce the number of H-bond donors, which is a key factor determining the sensitivity of nitrate-CIMS (Shen et al., 2022; Hyttinen et al., 2015). Recently, Li et al. (2024) used CI-Orbitrap with ammonium or nitrate reagent ions to detect oxygenated organic molecules in the synergistic $O_3$ + $NO_3$ regime and found that both the ion intensity of ONs and their signal contribution to the total dimers were much lower when using nitrate as reagent ions.

We have added the above paragraph to Section 2.1 of the revised manuscript.

4. Line 185-189. Besides these two reasons, it is also possible that the presence of $NO_2$ scavenged all acyl $RO_2$, which may be key to forming dimers. Earlier in the text, the authors stated that $RO_2$ + $NO_2$ reactions are considered. How about acyl$RO_2$ + $NO_2$ specifically to remove acyl$RO_2$s out of the system? $^{CI}RO_2$ contain more aldehydes and thus its product $RO_2$s are more likely acyl$RO_2$ than the $^{OH}RO_2$. This could make sense if $NO_2$ has a major impact on the termination reactions for the $^{CI}RO_2$ pathways.

Response: We appreciate the reviewer's point. The model simulations show that the concentrations of acyl $RO_2$ decrease by 60 – 79 % due to the consumption by $NO_2$. However, our previous study found that in the absence of NO, acyl $RO_2$ accounts for a significant fraction (32 – 94%) of $C_7$ – $C_9$ $RO_2$ but a very small fraction (0.4%) of $C_{10}$ $RO_2$ (Zang et al., 2023). As the α-pinene HOMs are dominated by $C_{10}$ species, the consumption of acyl $RO_2$ by $NO_2$ only leads to reductions of 4 – 5 % and 7 – 12 % in total

$C_xH_yO_z$-HOM monomers and dimers, respectively. Therefore, the significant reduction in $C_xH_yO_z$-HOMs in the synergistic oxidation regime is primarily due to the cross reactions of $^{Cl}RO_2$ and $^{OH}RO_2$ with $^{NO3}RO_2$. In addition, because of the very small contribution of acyl $RO_2$ to total $C_{10}$ $RO_2$, their consumption by $NO_2$ leads to less than 2% reduction in the $C_{10}$ $^{Cl}RO_2$ signals, and the larger decrease in $^{Cl}RO_2$ and related HOMs as compared to the OH-derived ones is mainly due to the more efficient cross reactions of $^{NO3}RO_2$ with $^{Cl}RO_2$ than with $^{OH}RO_2$.

We have added a discussion of the effect of $NO_2$ in Section 3.1.

"Meanwhile, the depletion of acyl $RO_2$ by $NO_2$ only leads to a small reduction (4 – 5% and 7 – 12%, respectively) in total $C_xH_yO_z$-HOM monomers and dimers in the synergistic regime compared to the $O_3$-only regime."

We have also added a discussion of the effect of $NO_2$ on the relative changes in $^{Cl}RO_2$ in Section 3.2.

"Because of the very small contribution of acyl $RO_2$ to the total $C_{10}$ $RO_2$ (0.4%) (Zang et al., 2023), their consumption by $NO_2$ leads to less than 2% reduction in the $C_{10}$ $^{Cl}RO_2$ signals. Therefore, the more significant decrease in signals of $^{Cl}RO_2$ and related HOMs as compared to the OH-derived ones in the synergistic $O_3 + NO_3$ regime is primarily due to the more efficient cross reactions of $^{NO3}RO_2$ with $^{Cl}RO_2$ than with $^{OH}RO_2$."

5. Figure 1. For (a) and (b), I suggest further clarifying what fractions of the $RO_2$, monomers, and dimers are made of compounds containing nitrogen. For (c), I suggest including CHO compounds as well, but using a different color. It might be also nice to show a mass spectrum with $O_3$ only, so that the comparison can be more clarified. In Line 207, the authors claimed "substantial formation of these dimeric ONs"; having a direct comparison can support this. In (c), $C_{10}H_{17}NO_8$ is the largest peak. Its formation should be briefly discussed. How does it form if $C_{10}H_{16}NO_5$ does not autoxidize rapidly, and the RO from $RO_2+RO_2$ reactions mainly release $NO_2$ and produce pinonaldehyde? Besides these suggestions, I wonder if the relative changes can be affected if the sensitivities are different from different species. This is such a major assumption, but it was not discussed in the manuscript.

Response: Thanks for the reviewer's comment.

(1) We have replaced Figure 1a with a new figure that shows the signals of total $RO_2$, total monomers, and total dimers normalized by the total reacted α-pinene, with the bars subdivided to indicate the fractions of CHO and CHON species, in both $O_3$-only and $O_3 + NO_3$ systems. In addition, we have provided a difference mass spectrum (i.e., mass spectrum in $O_3 + NO_3$ regime minus that in $O_3$-only regime) in Figure 1c, which highlights the changes in the species distribution in the synergistic oxidation regime compared to the $O_3$-only regime.

We have rewritten the discussion of this figure in Section 3.1 of the revised manuscript.

"The abundance of gas-phase $RO_2$ species and HOMs in different oxidation regimes is shown in Figure 1a. The species signals are normalized by the total reacted α-pinene in each regime. Compared to the $O_3$-only regime, the normalized signals of total $RO_2$ and HOMs decrease by 62 – 68% in the synergistic $O_3 + NO_3$ regime. Although $NO_3$ oxidation accounts for a considerable fraction of reacted α-pinene in the synergetic oxidation regime, the signal contributions of HOM-ONs are not significant. This might be due to the low sensitivity of nitrate-CIMS to the ONs formed involving $NO_3$ oxidation (Section 2.1). ……Figure 1c shows a difference mass spectrum highlighting the changes in species distribution

between the two oxidation regimes. Almost all $C_xH_yO_z$-HOM species decrease significantly in the $O_3$ + $NO_3$ regime compared to the $O_3$-only regime. Besides, a large set of HOM-ON species are formed, despite their relatively low signals.......”

[Figure]

Figure 1 Distributions of $RO_2$ and HOMs in the $O_3$-only and $O_3$ + $NO_3$ regimes. (a) Signals of total $RO_2$, as well as HOM monomers and dimers normalized by the reacted α-pinene in each oxidation regime (Exps 1-5, 7-11). (b) Relative changes in the normalized signals of $C_xH_yO_z$-HOMs in the $O_3$ + $NO_3$ regime versus the $O_3$-only regime. Ion signals are normalized to $\Delta[\text{α-pinene}]_{O3}$ in each oxidation regime to highlight the suppression effect of the synergistic chemistry between $^{NO3}RO_2$ and $^{Cl}RO_2$ or $^{OH}RO_2$ on $C_xH_yO_z$-HOM formation. (c) Difference mass spectrum between the two oxidation regimes. The positive and negative peaks indicate the species with enhanced and decreased formation in the $O_3$ + $NO_3$ regime compared to the $O_3$-only regime, respectively.

(2) There are two possible explanations for the relatively high signal intensity of $C_{10}H_{17}NO_8$: (i) Although the RO radicals from cross reactions of $C_{10}H_{16}NO_5$-$RO_2$ are prone to release $NO_2$ and form pinonaldehyde, a small fraction of them possibly undergo intramolecular H-shift/$O_2$ addition to form $C_{10}H_{16}NO_6$-$RO_2$, followed by further autoxidation to form $C_{10}H_{17}NO_8$; (ii) Although CI is a soft ionization method, the fragmentation of chemically labile species still occurs during the ionization in nitrate-CIMS. It is possible that some of dimeric HOM-ONs are fragmented to $C_{10}H_{17}NO_8$ during nitrate-CIMS measurements. We noticed that in a recent study by Li et al. (2014), $C_{10}H_{17}NO_8$ was also identified

during the synergistic oxidation of α-pinene by $O_3$ and $NO_3$. However, the exact origin of this species remains to be clarified.

We have rewritten this part of discussion in the revised manuscript.

"As shown in Figure 1c, although several closed-shell monomeric HOM-ONs have been observed in the synergistic oxidation regime, only a few of them exhibit relatively high signals. Among them, $C_{10}H_{17}NO_8$ may be formed by the autoxidation of $C_{10}H_{16}NO_6$-$RO_2$ derived from the intramolecular H-shift of primary $^{NO3}RO$ radicals ($C_{10}H_{16}NO_4$-RO). In addition, although CI is a soft ionization method, the fragmentation of chemically labile species still occurs during the ionization in nitrate-CIMS. It is possible that some of dimeric HOM-ONs are fragmented to $C_{10}H_{17}NO_8$ during nitrate-CIMS measurements. In a recent study by Li et al. (2024), $C_{10}H_{17}NO_8$ was also identified during the synergistic oxidation of α-pinene by $O_3$ and $NO_3$. However, the exact origin of this species remains to be clarified."

(3) Considering that different compounds could potentially have different CIMS sensitivities, we have conducted a sensitivity analysis by using different instrument sensitivities for different compounds to clarify their influences on the relative changes in $RO_2$ and HOMs in the $O_3$ + $NO_3$ regime versus the $O_3$-only regime. Taking a 10 times higher sensitivity to the compounds with an O/C ratio less than 0.7, the total signals are elevated in both oxidation regimes, but there remain significant decreases in total $RO_2$ and HOM signals in the synergistic oxidation regime compared to the $O_3$-only regime (Figure S4a). In addition, given that the sensitivity of nitrate-CIMS to ONs are relatively low, a 10 times higher sensitivity was also considered for the ONs. Under this condition, although ONs make a larger contribution to the total HOM monomers and dimers in the $O_3$ + $NO_3$ regime (Figure S4b), the signals of both total and $C_xH_yO_z$ $RO_2$ and HOMs still decrease significantly due to the presence of $NO_3$ oxidation. Therefore, different instrument sensitivities to $RO_2$ and HOMs with different oxygenation levels would not significantly influence the results (e.g., Figure 1) in this study.

[Figure]

Figure S4 Influences of different instrument sensitivities on the relative changes in $RO_2$ and HOMs in the synergistic oxidation regime versus the $O_3$-only regime. A 10 times higher instrument sensitivity to (a) compounds with O/C < 0.7 and (b) ONs was considered.

We have added the above discussion in Section S1 of the supplement and the following statement in Section 3.1 of the main text.

"Although there remain considerable uncertainties in instrument sensitivities to different compounds,

sensitivity analyses suggest that varying the CIMS sensitivities to $RO_2$ and HOMs by a factor of 10 would not significantly influence their relative distribution across different oxidation regimes (see Section S1 for details)."

6. Line 249. This is related to comment #1. It is not true if the different $RO_2$ cross reactions could also change branching ratios of ROOR. This possibility needs to be discussed.

Response: Thanks for the reviewer's comment. Recent studies suggested that the rate constants of the ROOR dimer formation for the highly oxygenated $RO_2$ appear to be fast (Berndt et al., 2018; Molteni et al., 2019), therefore a relatively high dimer formation branching ratio of 50% was used in this study. This branching ratio does not change with different $RO_2$ cross reactions. To estimate the influence of dimer formation branching ratio on the relative changes in $RO_2$ and related HOM concentrations in the synergistic $O_3 + NO_3$ regime versus the $O_3$-only regime, we have conducted a sensitivity analysis of this ratio and added the following discussion to Section S3 in the Supplement.

"Currently, quantitative constraints on the ROOR dimer formation rate constant are rather limited. Recent studies suggested that the dimer formation rates from the highly oxygenated $RO_2$ are fast (Berndt et al., 2018; Molteni et al., 2019), therefore a relatively high and consistent dimer formation branching ratio of 50% was used for different $RO_2$ (e.g., $^{Cl}RO_2$, $^{OH}RO_2$, $^{NO3}RO_2$) in this study. Considering the large uncertainties in this branching ratio, we conducted a sensitivity analysis to evaluate its influence on the relative changes in $RO_2$ and related HOM concentrations in the synergistic $O_3 + NO_3$ regime versus the $O_3$-only regime. As shown in Figure S8, as the dimer formation branching ratio increases from 9% to 50%, the variation in the abundance $C_xH_yO_z$-$RO_2$ and HOMs due to the concurrence of $NO_3$ oxidation changes slightly (< 9% and < 10%, respectively). These results suggest that the uncertainties in the dimer formation branching ratio of $RO_2$ cross-reactions do not significantly affect the distribution of $RO_2$ and HOMs across different oxidation regimes."

[Figure]

Figure S8 Influences of the dimer formation branching ratio on the relative changes in $RO_2$ and related HOM concentrations in the synergistic $O_3 + NO_3$ regime vs. the $O_3$-only regime.

7. Line 290-294. Can these findings be explained by the kinetic model?

Response: The model simulations show that the concentrations of $C_{20}H_{31}NO_x$ in the $O_3 + NO_3$ regime increase with the addition of cyclohexane as an OH scavenger (Figure S9). However, the simulated enhancement is slightly lower than the measurements, which might be due to the uncertainties in the $RO_2$ cross-reaction kinetics in the model.

We have added the above discussion to Section 3.2 of the main text and Figure S9 to the Supplement.

[Figure]

Figure S9 Simulated and measured relative changes in concentrations of $C_{20}H_{31}NO_x$ due to the addition of 100 ppm cyclohexane as an OH scavenger in the synergistic $O_3 + NO_3$ regime (Exps 6 and 12).

8. Line 326-328. However, the $C^*$ distribution in Figure 5 does not show higher abundance for the SVOC & IVOC range under $NO_3/O_3$ mixed oxidation conditions. How come the SOA mass loading is higher?

Response: The higher SOA mass loading in the synergistic oxidation regime is mainly due to the formation and condensation of HOM-ONs. However, as discussed in our response to comment #3, nitrate-CIMS may exhibit a significantly lower sensitivity to ONs than to $C_xH_yO_z$-HOMs, thus the measured signals of HOM-ONs are relatively low and have a small contribution to SVOC signals in the $O_3 + NO_3$ regime.

We have added a more detailed discussion regarding the growth of particles in Section 3.3.

"On the other hand, substantial formation of HOM-ONs is expected from the cross reactions of $^{NO3}RO_2$ with $^{Cl}RO_2$ and $^{OH}RO_2$ in the synergistic oxidation regime (Li et al., 2024; Bates et al., 2022), although their signals are relatively low due to the low sensitivity of nitrate-CIMS to ONs in this study. The newly formed HOM-ONs have relatively higher volatilities and are inefficient in initiating particle nucleation, but they are able to partition into the formed particles and contribute to the particle mass growth. Meanwhile, as the particle number concentration decreases drastically in the synergistic oxidation regime, more condensable vapors are available for each particle to grow to larger sizes (Figure 6b), which would in turn favor the condensation of more volatile organic species including ONs due to the reduced curvature effect of the larger particles, ultimately resulting in an increase in SOA mass concentrations."

[Figure]

Figure 6b. Size distributions of particles formed from the ozonolysis and synergistic $O_3 + NO_3$ oxidation of α-pinene (Exps 13-14).

9. Section 3.4. It is nice to expand the chemistry into real-world conditions. The authors considered boreal forest conditions where monoterpenes are high. But they also mentioned southeast US conditions, where isoprene is high. Can the southeast US scenario be modeled? I think this is doable as the same authors published a paper on mixed isoprene/monoterpene oxidation.

Response: We appreciate the review's point. We have conducted a model simulation to evaluate the influence of the synergistic $O_3 + NO_3$ oxidation on HOM formation under typical nocturnal conditions in the southeastern US. We find that in the mixed isoprene/monoterpene oxidation regime, the synergistic $O_3 + NO_3$ oxidation can still suppress the formation of $C_xH_yO_z$-HOM monomers and dimers, and the presence of isoprene can strengthen this inhibition effect to some extent.

We have added the following detailed discussion to Section S4 of the Supplement.

"A model simulation was also conducted to evaluate the influences of synergistic oxidation on HOM formation under typical nocturnal conditions of the southeastern United States. The constant concentrations of α-pinene (1.5 ppb), isoprene (4.5 ppb), $O_3$ (30 ppb), NO (20 ppt), $NO_2$ (2 ppb), $NO_3$ radicals (1.4 ppt), OH radicals ($2.5 \times 10^5$ molecules $cm^{-3}$), and $HO_2$ radicals (4 ppt) were used according to field observations in this region (Ayres et al., 2015; Lee et al., 2016). The rate constant of self/cross reactions involving isoprene-derived $RO_2$ radicals (termed $RO_2$(isop)) was set to $2 \times 10^{-12}$ $cm^3$ molecule$^{-1}$ $s^{-1}$, with a dimer formation branching ratio of 50% for $RO_2$(isop) with $RO_2$ arising from α-pinene (termed $RO_2$(αp)) and 30% for $RO_2$(isop) + for $RO_2$(isop) (Berndt et al., 2018; Wang et al., 2021)."

We have also added the following paragraph to Section 3.4 of the main text.

"Furthermore, model simulations for conditions typical of the southeastern United States (see details in Section S4) suggest that the coexistence of isoprene appears to exacerbate the suppression effect of synergistic oxidation on HOM formation from monoterpenes. As shown in Figure S14, in the absence of isoprene, the synergistic $O_3 + NO_3$ oxidation of α-pinene leads to a reduction of 13% and 24% in the formation of $C_xH_yO_z$-HOM monomers and dimers, respectively. When isoprene is present, as the isoprene + $NO_3$ oxidation produces a significant amount of nitrooxy $RO_2$ that can aslo scavenge α-pinene-derived $^{Cl}RO_2$ and $^{OH}RO_2$ via cross reactions, the synergistic oxidation leads to a slightly larger reduction in $C_xH_yO_z$-HOM monomers and dimers (15% and 31%, respectively)."

[Figure]

Figure S14 Simulated concentrations of $C_xH_yO_z$-HOMs from the ozonolysis and synergistic $O_3 + NO_3$ oxidation of α-pinene in the (a) absence and (b) presence of isoprene under typical nocturnal conditions in the southeastern United States.

Technical comments:

1. Line 109. Change "their" to "its".

Response: We have revised this.

2. Line 164. Does $^{NO3}RO_2$ represent only the primary $RO_2$ from $NO_3$ + alpha-pinene (i.e., $C_{10}H_{16}O_5$-$RO_2$) throughout the manuscript? It should be clarified if that is the case.

Response: $^{NO3}RO_2$ represents the $RO_2$ radicals from $NO_3$ + α-pinene, and here we want to highlight the primary $^{NO3}RO_2$, i.e., $C_{10}H_{16}NO_5$-$RO_2$, the cross reactions of which were added to the model.

We have clarified this statement in the revised manuscript.

"We added the cross reactions of the primary nitrooxy-$RO_2$ derived from $NO_3$ oxidation ($^{NO3}RO_2$), i.e., $C_{10}H_{16}NO_5$-$RO_2$, with $RO_2$ derived from ozonolysis ($^{CI}RO_2$) and OH oxidation ($^{OH}RO_2$)."

3. Line 233. Change "strong" to "stronger".

Response: We have revised this.

4. Figure 4. On the y-axis, "conc" is not accurate. Should be intensity or signal. Also, does "CA" mean cyclohexane? It should be clarified.

Response: We have changed "conc" to "signal" and "CA" to "cyclohexane".

References

[revised manuscript text omitted]

---

## Author Comment (AC2)

We are grateful to the reviewer for the thoughtful comments on the manuscript. Our point-to-point responses to each comment are as follows (reviewer's comments are in black font and our responses are in blue font).

General Comments

Zang and coworkers investigated synergistic effects on the reduction of low-volatile organic compounds during nighttime oxidation of a-pinene. Through laboratory flow tube experiments, the authors found that $NO_3$-$RO_2$ reacts with CI-$RO_2$/OH-$RO_2$ and impedes the formation of low-volatile HOMs that would form a secondary organic aerosol. The results robustly show the synergistic effect on low-volatile organic compound reduction via well-designed experiments under conditions with and without NO3 radicals. The findings in this study would improve our understanding of complex and more realistic environments where different atmospheric radicals present and affect the oxidation chemistry of biogenic volatile organic compounds.

However, there are drawbacks in this study that need to be improved. My main concern is that the experimental conditions would not successfully represent the ambient atmosphere conditions. In Section 3.4., the authors commented on the input conditions of the model they ran, which were similar to the ambient atmosphere conditions of boreal forests reported in previous studies. While the authors ran the model under humid conditions, lab experiments in this study were performed only under dry conditions. The humidity condition would affect $RO_2$/ozonolysis reaction chemistry as well as the fate of Criegee intermediates and the other oxidation products. I suggest conducting additional experiments and validating if the authors would get the same experimental results between dry and humid conditions, and then applying such results to the model to understand if the findings in this study can be applied to the actual ambient environment.

Response: Thanks for the reviewer's comment. We agree that under humid conditions, water vapor may affect the fate of Criegee intermediates (CIs) and $RO_2$ radicals and thereby product formation during the oxidation of organics. However, the importance of such effects is highly dependent on the molecular size and structure of the precursor organics. Overall, in the α-pinene oxidation system, the influences of RH on the chemistry of CIs and $RO_2$, as well as the formation of HOMs are small (see details below).

Kinetics studies have found that the stabilized Criegee intermediates (SCIs) arising from α-pinene ozonolysis can undergo fast unimolecular decay at a rate constant of $60 - 250$ s$^{-1}$ (Vereecken et al., 2017; Newland et al., 2018), which is rapid compared to their reaction with water vapor, in particular for syn-SCIs, under atmospheric conditions (Vereecken et al., 2017; Newland et al., 2018). In addition, the yield of OH radicals from CI decomposition is independent of RH (Atkinson et al., 1992; Aschmann et al., 2002). Consistent with the fast unimolecular reaction kinetics revealed by these studies, recent laboratory measurements have shown that the contribution of SCIs to the formation of gas-phase and particle-phase dimers are small (<20%) during α-pinene ozonolysis (Zhao et al., 2018; Zhao et al., 2022). Furthermore, the molecular composition and abundance of HOM monomers and dimers (Li et al., 2019) and the formation of particle-phase dimers (Zhang et al., 2015; Kenseth et al., 2018) do not change significantly with RH ranging from 3% to 92%. These studies suggest that the humidity condition does not strongly affect the HOM formation chemistry in the α-pinene ozonolysis system. In the present study, using a kinetic model updated with the latest advances in the $RO_2$ and CI chemistry, we also find a large decrease in $C_xH_yO_z$-HOMs due to the synergistic $O_3$ + $NO_3$ oxidation under typical nocturnal atmospheric

conditions (RH = 50%), demonstrating that the conclusions obtained from the flow tube experiments are also valid under typical atmospheric conditions. In addition, model simulations show that the variation in RH has negligible influence on the relative changes in $C_xH_yO_z$-HOMs under typical nocturnal atmospheric conditions (Figure S11).

[Figure]

Figure S11 Influence of relative humidity on the relative changes of $C_xH_yO_z$-HOMs in the $O_3 + NO_3$ regime compared to those in the $O_3$-only regime under typical nocturnal atmospheric conditions.

We have added the above discussion to Section 3.4 of the revised manuscript.

Specific Comments

Line 180 - 183: What was the $RO_2$ fate like at each experiment? Might be helpful if providing figures in the SI

Response: Thanks for the reviewer's comment. We have added a figure showing the $RO_2$ fate in both $O_3$-only and $O_3 + NO_3$ regimes in the SI. We have also added the following statement in Section 3.1.

"Also, $NO_3$ radicals almost entirely (over 98.5%) react with α-pinene and their reaction with $RO_2$ has negligible influence on the fate of $RO_2$ (Figure S2).

[Figure]

Figure S2 $RO_2$ fates in the (a) $O_3$-only and (b, c) $O_3 + NO_3$ regimes, taking $C_{10}H_{15}O_6$-$^{Cl}RO_2$ in Exps 3 and 8 as an example. The reactions of $NO_3 + RO_2$ are considered in (b) but not in (c)."

Line 192: Why did you normalize by $\Delta$[a-pinene]$O_3$? Please add a more detailed explanation.

Response: There are two major reasons for the strong reduction in HOM formation in the synergistic oxidation regime compared to the $O_3$-only regime: (i) the fast competitive consumption of α-pinene by $NO_3$ radicals, which leads to a reduction in the reacted α-pinene by $O_3$ ($\Delta$[α-pinene]$_{O_3}$) and thereby $C_xH_yO_z$-HOM signals, and (ii) the cross reactions of $^{NO3}RO_2$ with $^{Cl}RO_2$ or $^{OH}RO_2$, which suppress the

autoxidation and self/cross reactions of $^{Cl}RO_2$ and $^{OH}RO_2$ to form $C_xH_yO_z$-HOMs. To quantify the contribution of synergistic cross reactions of $^{NO3}RO_2$ with $^{Cl}RO_2/^{OH}RO_2$ to the suppressed formation of $C_xH_yO_z$-HOMs in the synergistic oxidation regime, $C_xH_yO_z$-HOM signals are first normalized to $\Delta[\alpha$-pinene$]_{O3}$ in each oxidation regime and then compared between different oxidation regimes.

We have added the following explanations in the revised manuscript.

"The strong reduction in HOM formation in the synergistic oxidation regime compared to the $O_3$-only regime is likely due to (i) the fast competitive consumption of α-pinene by $NO_3$ radicals, which leads to a reduction in the reacted α-pinene by $O_3$ ($\Delta[\alpha$-pinene$]_{O3}$, Figure S3) and thereby $C_xH_yO_z$-HOM signals, and (ii) the cross reactions of $^{Cl}RO_2$ or $^{OH}RO_2$ with $^{NO3}RO_2$, which suppress the autoxidation and self/cross reactions of $^{Cl}RO_2$ and $^{OH}RO_2$ to form $C_xH_yO_z$-HOMs. To quantify the contribution of synergistic cross reactions of $^{NO3}RO_2$ with $^{Cl}RO_2/^{OH}RO_2$ to the suppressed formation of $C_xH_yO_z$-HOMs in the synergistic oxidation regime, $C_xH_yO_z$-HOM signals shown in Figure 1a are first normalized to $\Delta[\alpha$-pinene$]_{O3}$ in each oxidation regime and then compared between different oxidation regimes (see Figure 1b)."

Line 209 - 213: Would the low signal of $NO_3$-$RO_2$ ($C_{10}H_{16}NO_x$) be indeed because of less autoxidation? Or could it be due to $NO_3$-CIMS's limitation on sensitivity over such compounds? Were there possibilities that unidentified compounds were being lost to the wall or particles?

Response: We appreciate the reviewer's point. The low signals of $^{NO3}RO_2$ are partly due to the relatively low sensitivity of nitrate-CIMS to such compounds. In addition, the instrument's mass resolution is not high enough to differentiate the mass closure between some of $^{NO3}RO_2$ and $C_xH_yO_z$-HOMs, limiting the detection of $^{NO3}RO_2$ species. During the HOM formation experiments, there was extremely low SOA formation observed by SMPS (see details in our responses to the reviewer #1, comments #2), which would have negligible effects on the production and signals of $^{NO3}RO_2$ species. In addition, the model simulation shows that wall losses only account for $7 - 8\%$ of the total production of $^{NO3}RO_2$ under various experimental conditions.

We have revised the explanations for low $^{NO3}RO_2$ signals in the revised manuscript.

"It should be noted that no obvious signals of highly oxygenated $^{NO3}RO_2$ ($C_{10}H_{16}NO_x$, $x \geq 6$) were observed by nitrate-CIMS in the synergistic $O_3 + NO_3$ oxidation system. One possible reason is that nitrate-CIMS exhibits relatively low sensitivity to the organic nitrates. Secondly, the instrument's mass resolution is not high enough to differentiate the mass closure between some of $^{NO3}RO_2$ and $C_xH_yO_z$-HOMs with strong peaks (Table S3), limiting the detection of $^{NO3}RO_2$ species. In addition, previous studies revealed that the primary $^{NO3}RO_2$ radicals (i.e., $C_{10}H_{16}NO_5$-$RO_2$) in the α-pinene + $NO_3$ system mainly react to form pinonaldehyde (Kurtén et al., 2017; Perraud et al., 2010)……"

Line 225: What are the "other reactions" in Figure S3? Please specify in the legend or embed those reactions in the figure. Also, would H-abstraction by $NO_3$ be small?

Response: "Other reactions" are α-pinene ozonolysis and OH oxidation by addition, we have clarified it in the legend.

We have added a discussion about H-abstraction by $NO_3$ in Section 3.2 of the revised manuscript.

"However, during the $NO_3$ oxidation of monoterpene, the rate constant for H-abstraction by $NO_3$ radicals

is $(4 - 10) \times 10^{-17}$ cm$^3$ molecule$^{-1}$ s$^{-1}$, which is $10^3 - 10^4$ lower than the rate constant for the NO$_3$ addition channel (Martinez et al., 1998). Besides, the subsequent reactions of RO$_2$ species formed from H-abstraction by NO$_3$ radicals should be very similar to those derived from H-abstraction by OH radicals, which was found not important for C$_x$H$_y$O$_z$-HOM formation in the absence of NO (Figure S5). Therefore, the H-abstraction of α-pinene by NO$_3$ radicals would have negligible influence on C$_{10}$H$_{15}$O$_x$ formation."

Line 233: Do you expect the predominant type of RO$_2$ would be different among CI-RO$_2$, NO$_3$-RO$_2$, and OH-RO$_2$ (i.e. if they are primary, secondary, tertiary, or acyl-RO$_2$)? Could you add more discussion on the NO$_3$-RO$_2$'s termination effect?

Response: Thanks for the reviewer's comment. The second-generation oxidation processes are strongly inhibited due to an excess of α-pinene in this study. As a result, the predominant type of RO$_2$ observed is primary RO$_2$. Our previous study found that in the absence of NO, acyl RO$_2$ contributes to a significant fraction of C$_7$ – C$_9$ RO$_2$, but a very small fraction of C$_{10}$ RO$_2$ (Zang et al., 2023). In the present study, the model simulations show that the consumption of acyl RO$_2$ by NO$_2$ lead to reductions of 4 – 5 % and 7 – 12 % in the total C$_x$H$_y$O$_z$-HOM monomer and dimers, respectively. Therefore, the significant reduction in C$_x$H$_y$O$_z$-HOMs in the synergistic oxidation regime is primarily due to the cross reactions of $^{NO3}$RO$_2$ with $^{CI}$RO$_2$ or $^{OH}$RO$_2$. In addition, because of the very small contribution of acyl RO$_2$ to the total C$_{10}$ RO$_2$, their consumption by NO$_2$ leads to less than 2% reduction in the $^{CI}$RO$_2$ signals, and the larger decrease in $^{CI}$RO$_2$ and related HOMs as compared to the OH-derived ones is mainly due to the more efficient cross reactions of $^{NO3}$RO$_2$ with $^{CI}$RO$_2$ than with $^{OH}$RO$_2$.

We have added the discussion of the type of RO$_2$ in Section 3.2.

"It should be noted that the second-generation oxidation processes are strongly inhibited by the excess of α-pinene in this study, thus the predominant type of RO$_2$ observed is primary RO$_2$."

In addition, we have added additional discussion regarding the effect of acyl RO$_2$ consumption by NO$_2$ and the $^{NO3}$RO$_2$'s termination effect in Sections 3.1 and 3.2.

Section 3.1: "Meanwhile, the depletion of acyl RO$_2$ by NO$_2$ only leads to a small reduction (4 – 5% and 7 – 12%, respectively) in total C$_x$H$_y$O$_z$-HOM monomers and dimers in the synergistic regime compared to the O$_3$-only regime."

Section 3.2: "Because of the very small contribution of acyl RO$_2$ to the total C$_{10}$ RO$_2$ (0.4%) (Zang et al., 2023), their consumption by NO$_2$ leads to less than 2% reduction in the C$_{10}$ $^{CI}$RO$_2$ signals. Therefore, the more significant decrease in signals of $^{CI}$RO$_2$ and related HOMs as compared to the OH-derived ones in the synergistic O$_3$ + NO$_3$ regime is primarily due to the more efficient cross reactions of $^{NO3}$RO$_2$ with $^{CI}$RO$_2$ than with $^{OH}$RO$_2$."

Cyclohexane experiment: Why haven't you run any SOA experiments for this condition? If this experiment was just for a sanity check, I suggest moving it to SI. Also, is there a reason why some of OH-RO$_2$ and HOMs monomer species in Figure 2 are not shown in Figure 4 (i.e., C$_{10}$H$_{17}$O$_{10}$, C$_{10}$H$_{18}$O$_{11}$)? Additionally, if you labeled specific ON-HOM compounds in Figure 1c, you should have shown how they changed in Figure 4c as well.

Response: The presence of cyclohexane could also affect the SOA formation and composition. But in the present study, we mainly focused on the gas-phase chemistry and did not run SOA experiments with the addition of cyclohexane.

We have added all RO₂, HOMs, and HOM-ONs shown in Figures 1c and 2 to Figure 4.

[Figure]

Figure 4. Relative changes in signals of (a) $C_{10}$ RO₂, (b) $C_{10}$ HOMs, and (c) $C_{20}$ dimers due to the addition of 100 ppm cyclohexane as an OH scavenger in the synergistic $O_3$ + NO₃ regime (Exps 7 and 12)."

Line 301: Weren't the results up to this line showing that CHO-HOMs were terminated via NO₃-RO₂ and CI-RO₂ reactions? Little via OH-RO₂?

Response: Thanks, we have rewritten this sentence in the revised manuscript.

"Compared to the $O_3$-only regime, there are a remarkable reduction in $C_xH_yO_z$-HOMs and a strong formation of HOM-ONs, which is mainly due to the efficient cross reactions between $^{NO3}RO_2$ and $^{Cl}RO_2$ in the synergistic oxidation regime."

Line 328: I suggest the authors shall add more discussion on particle formation and growth. What is the main factor that drives larger mass SOA concentration? Did you identify more numbers of compounds showing higher signals over certain thresholds? Was the entire sum of CPS different by reaction conditions?

Response: Thanks for the reviewer's comment.

1. We have added more discussion regarding the particle formation and growth and compared the results with the latest studies in Section 3.3 of the main text.

"Figure 6a shows the particle number and mass concentrations formed in the two oxidation regimes in the SOA formation experiments (Table S1, Exps 13 and 14). The particle number concentration decreases by more than 50% whereas the particle mass concentration increases by a factor of 2 in the synergistic

$O_3$ + $NO_3$ regime, compared to that in the $O_3$-only regime. The presence of $NO_3$ radicals during α-pinene ozonolysis significantly reduces the abundance of ULVOCs and ELVOCs, which are the key species driving particle nucleation (Simon et al., 2020; Schervish and Donahue, 2020), thereby leading to a large reduction in the particle number concentration in the synergistic $O_3$ + $NO_3$ regime.

On the other hand, substantial formation of HOM-ONs is expected from the cross reactions of $^{NO3}RO_2$ with $^{Cl}RO_2$ and $^{OH}RO_2$ in the synergistic oxidation regime (Li et al., 2024; Bates et al., 2022), although their signals are relatively low due to the low sensitivity of nitrate-CIMS to ONs in this study. The newly formed HOM-ONs have relatively higher volatilities and are inefficient in initiating particle nucleation, but they are able to partition into the formed particles and contribute to the particle mass growth. Meanwhile, as the particle number concentration decreases drastically in the synergistic oxidation regime, more condensable vapors are available for each particle to grow to larger sizes (Figure 6b), which would in turn favor the condensation of more volatile organic species including ONs due to the reduced curvature effect of the larger particles, ultimately resulting in an increase in SOA mass concentrations. Recently, Bates et al. (2022) also found that in chamber experiments with seed particles, the SOA mass yields were significantly higher during α-pinene oxidation by $O_3$ + $NO_3$ than during ozonolysis, mainly due to the substantial formation and condensation of ON dimers."

[Figure]

Figure 6. Number and mass concentrations (a), as well as the size distribution (b) of particles formed from the ozonolysis and synergistic $O_3$ + $NO_3$ oxidation of α-pinene (Exps 13-14).

2. The sum of CPS is very similar under different reaction conditions, i.e., (5.9 – 6.2) × $10^4$ in both oxidation regimes. We have added this information in section 2.1.

"The total ion counts (TIC) with values of (5.9 – 6.2) × $10^4$ cps are similar under different reaction conditions."

Figure 6: At least in SI, I would like to see how size distribution is different between the experimental conditions, and how they vary. That comparison may give some insights into the observation in Figure 6.

Response: Thanks for the reviewer's comment. As described in our responses to last comment, we have added a figure showing the particle size distributions in different oxidation regimes as well as the relevant discussions to the revised manuscript.

Line 338: How well do the experiments reflect the given ambient condition? How were NO and $NO_2$ concentrations in the experiments? How would RH variation affect $NO_3$/$N_2O_5$? How would the aqueous-phase reaction affect $RO_2$ formation and fate? Also, high RH would have hydrolysis of ON-HOMs and

the reaction mechanism/products would not be the same as what you explored in your experiments. I think you should validate from additional humid condition experiments if your experimental results can be applied to the atmospheric models regardless of the humidity conditions.

Response: Thanks for the reviewer's comment. In the present study, we aim to elucidate the role of synergistic $O_3$ and $NO_3$ oxidation of α-pinene in the formation of low-volatility organic compounds, in particular HOMs, under nighttime conditions. Therefore, we mainly focused on the characterization of the molecular composition and formation chemistry of gas-phase HOMs during the ozonolysis and synergistic $O_3$ + $NO_3$ oxidation of α-pinene using a combination of flow tube experiments (primarily with a short residence time to avoid significant production of particles) and detailed kinetic modelling (with the mechanism updated for the latest advances in $RO_2$ and CI chemistry). The initial $NO_2$ concentration in the flow tube was 4.5 ppb. To prevent the titration of $NO_3$ radicals by NO, all the experiments were performed without the addition of NO. To evaluate the atmospheric relevance of the experimental results, we have performed model simulations under typical nocturnal conditions in the boreal forest in Finland and in the southeastern US, for which representative concentrations of NO were considered. As discussed in our responses to the comment #1, the fates of $RO_2$ and CIs, as well as the formation of HOMs during α-pinene ozonolysis are not strongly affected by the RH. The main findings obtained from the flow tube experiments are corroborated by the model simulations under typical atmospheric conditions.

In this study, we have also performed a few SOA formation experiments to examine the effect of synergistic $NO_3$ + $O_3$ oxidation on the formation of SOA. We agree that under humid conditions, aerosol liquid water could affect the aging and composition of SOA, for example, by favoring the hydrolysis of particulate HOM-ONs. However, such processes would not change the main conclusions in the present study (e.g., the nocturnal synergistic $NO_3$ + $O_3$ oxidation significantly reduces the formation of ULVOCs from monoterpenes). Given the high abundance and lability of HOMs and HOM-ONs in SOA, the detailed composition and aging chemistry of SOA as well as the influence of RH warrant future investigations.

We have added the following experimental information to Section 2.1 of the revised manuscript.

"The initial $NO_2$ concentration in the flow tube was ~4.5 ppb. To prevent the titration of $NO_3$ radicals by NO, all the experiments were performed without the addition of NO."

We have added the following discussion to Section 3.4.

"In the present study, the flow tube experiments were conducted under dry conditions. Although water vapor may affect the fate of Criegee intermediates (CIs) and $RO_2$ radicals and thereby HOM formation during the oxidation of organics under humid conditions, there is growing evidence that such effects in the α-pinene oxidation system are small. Kinetics studies have found that the stabilized Criegee intermediates (SCIs) arising from α-pinene ozonolysis can undergo fast unimolecular decay at a rate constant of $60 - 250$ s$^{-1}$ (Vereecken et al., 2017; Newland et al., 2018), which is rapid compared to their reaction with water vapor, in particular for syn-SCIs, under atmospheric conditions (Vereecken et al., 2017; Newland et al., 2018). In addition, the yield of OH radicals from CI decomposition is independent of RH (Atkinson et al., 1992; Aschmann et al., 2002). Consistent with the fast unimolecular reaction kinetics revealed by these studies, recent laboratory measurements have shown that the contribution of SCIs to the formation of gas-phase and particle-phase dimers are small (<20%) during α-pinene

ozonolysis (Zhao et al., 2018; Zhao et al., 2022). Furthermore, the molecular composition and abundance of HOM monomers and dimers (Li et al., 2019) and the formation of particle-phase dimers (Zhang et al., 2015; Kenseth et al., 2018) do not change significantly with RH ranging from 3% to 92%. These studies suggest that the humidity condition does not strongly affect the HOM formation chemistry in the α-pinene ozonolysis system."

Line 374: Do these HOM monomers and dimers have high numbers of oxygen as what you observed from the lab experiments?

Response: The HOM dimers in the model have high numbers of oxygen (up to15) as we observed in the flow tube experiments. But for the simulated HOM monomers, their oxygen numbers are no more than 11. This is because the formation of $RO_2$ with oxygen numbers higher than 11 was not considered in the model, due to the large uncertainty in the autoxidation rate constants of the highly oxygenated $RO_2$. In fact, the autoxidation rate of the highly oxygenated $RO_2$ is expected to be small given the significant decrease in the number of active sites for intramolecular H-abstraction in the molecule. As a result, the contribution of the most oxygenated HOMs to the total HOM monomers could be relatively small (Zhao et al., 2018; Claflin et al., 2018).

We have added a relevant discussion to Section 3.4 of the revised manuscript.

"The formation of $RO_2$ with oxygen numbers higher than 11 was not considered in the model, due to the large uncertainty in the autoxidation rate constants of the highly oxygenated $RO_2$. In fact, the autoxidation rate of the highly oxygenated $RO_2$ is expected to be small given the significant decrease in the number of active sites for intramolecular H-abstraction in the molecule. As a result, the contribution of the most oxygenated HOMs to the total HOM monomers could be relatively small (Zhao et al., 2018; Claflin et al., 2018)."

Line 388: How about under very low $NO_2$, $NO_3$, and $N_2O_5$ environments? Would $NO_3$ still suppress CHO-HOMs during nighttime?

Response: Thanks for the reviewer's comment. We have considered a relatively low $NO_3$ concentration of 0.2 ppt in this study. Under this condition, ozonolysis is the primary loss pathway of α-pinene (68%), and $NO_3$ oxidation contributes to 30% of α-pinene oxidation. The synergistic $O_3$ + $NO_3$ oxidation of α-pinene leads to a reduction of 3% and 13% in the formation of $C_xH_yO_z$-HOM monomers and dimers, respectively. To get a picture for the very low $NO_2$ and $NO_3$ conditions, we also performed model simulations with $NO_2$ and $NO_3$ concentrations of 0.7 ppb and 0.08 ppt (Zhang et al., 2018), respectively. The simulated result shows that the under this condition, the vast majority of α-pinene is oxidized by $O_3$ (87%), and $NO_3$ only contributes for 9.6%. As a result, the influence of synergistic oxidation of $NO_3$ and $O_3$ on the HOM formation is minor, with a reduction of 1.4% and 6% in the formation of $C_xH_yO_z$-HOM monomers and dimers, respectively. However, it should be noted that in many forested atmospheres, $NO_3$ radicals play an important role in the nocturnal oxidation of monoterpenes (Bianchi et al., 2017; Yan et al., 2016; Lee et al., 2018) and significant effects of synergistic $O_3$ + $NO_3$ oxidation on low-volatility organics formation are expected.

I think you should add a discussion on the role of $CI-RO_2$ on dimer & ULVOC formation as well. Additional discussion on this based on the comparison with previous studies would help readers learn about nighttime oxidation chemistry and would help emphasize why your findings are important.

Response: We appreciate the reviewer's point. We have added a discussion on the role of [Cl]RO₂ on dimer and ULVOC formation.

"The above model simulations suggest that under nocturnal atmospheric conditions with a very low NO₃ concentration, the RO₂ radical pool is dominated by [Cl]RO₂ and their self/cross reactions are a major contributor to ULVOCs such as the highly oxygenated C₂₀ dimers as observed in boreal forest (Bianchi et al., 2017). When the NO₃ concentration is high, the production of [NO3]RO₂ becomes significant and their cross reactions with [Cl]RO₂ would suppress the formation of ULVOCs."

Technical comments:

Line 178: Please add a more detailed explanation on the y-axis of Figure 1.

Response: We have redrawn Figure 1, and added more detailed explanations for the y-axis in the figure caption.

[Figure]

Figure 1 Distributions of RO₂ and HOMs in the O₃-only and O₃ + NO₃ regimes. (a) Signals of total RO₂, as well as HOM monomers and dimers normalized by the reacted α-pinene in each oxidation regime (Exps 1-5, 7-11). (b) Relative changes in the normalized signals of $C_xH_yO_z$-HOMs in the O₃ + NO₃ regime versus the O₃-only regime. Ion signals are normalized to Δ[α-pinene]$_{O3}$ in each oxidation regime to highlight the suppression effect of the synergistic chemistry between [NO3]RO₂ and [Cl]RO₂ or [OH]RO₂ on $C_xH_yO_z$-HOM formation. (c) Difference mass spectrum between the two oxidation regimes. The positive and negative peaks indicate the species with enhanced and decreased formation in the O₃ + NO₃ regime compared to the O₃-only regime, respectively.

Line 180 & Table S1: How about adding a footnote of experimental conditions that are compared to each other?

Response: We have added a footnote as follows.

"Exps 1-6 and 7-12 are HOM formation experiments in the $O_3$-only and $NO_3+O_3$ regimes, respectively, and Exps 13 and 14 are SOA formation experiments in the two oxidation regimes."

Line 198: Could you also specify that these monomers & dimers are CHO-HOMs? Because the next figure focuses on ON-HOMs, it would be better to make it clear to avoid confusion.

Response: We have redrawn Figure 1 and specified $C_xH_yO_z$-$RO_2$, HOM monomers and dimers in the figure caption.

Line 200: Add "among experiments with same initial a-pinene concentration" before "(Exps 1-10)"

Response: Thanks, we have added it.

Line 215: Were you trying to say that the instrument's resolution is not good enough to separate these? If so, I would say "the instrument's resolution is not enough to differentiate the mass closure between $NO_3$-$RO_2$ and CHO-HOMs (Table S3), limiting the detection of $NO_3$-$RO_2$ species."

Response: We have revised this sentence as "In addition, the instrument's resolution is not high enough to differentiate the mass closure between some of $^{NO3}RO_2$ and $C_xH_yO_z$-HOMs (Table S3), limiting the detection of $^{NO3}RO_2$ species."

Line 257: Please add a statement in general words and specify what this reaction efficiency means to the observations in Figure 1 and/or 2 results.

Response: We have added a statement as follows: "Therefore, we conclude that the cross-reaction rate constants of $^{NO3}RO_2 + {}^{Cl}RO_2$ are on average $10 - 100$ times larger than those for $^{NO3}RO_2 + {}^{OH}RO_2$. This different $RO_2$ cross-reaction efficiency is the main reason for the significantly larger decrease in the abundance of $^{Cl}RO_2$ and related HOMs as compared to the OH-derived ones (see Figure 2)."

Figure 4: What is "CA" on the right axis?

Response: We have replaced "CA" with "cyclohexane" in the revised manuscript.

Line 307 & 311: Figure 5 only has one figure, not any subfigures

Response: We have revised this.

Line 349: I think it would be better to have a pie chart showing $RO_2$ fate in SI (both from your experiments and model application)

Response: The $RO_2$ fates in the HOM-formation experiments are added in Figure S2 (see our responses above), and the $RO_2$ fates under typical atmospheric conditions are added in Figure S10.

We have added a description of the $RO_2$ fates under typical atmospheric conditions in the revised manuscript.

"When a relatively low $NO_3$ concentration (0.2 ppt) is considered, ...... the reactions of $RO_2 + HO_2$, $RO_2 + NO$, and $RO_2 + RO_2$ account for ~49%, ~27%, and ~24% of the total $RO_2$ fate, respectively

(Figure S10a). When the $NO_3$ concentration is as high as 1 ppt as reported in field studies …… the $RO_2 + RO_2$ reactions account for ~34% of the total $RO_2$ fate (Figure S10b)"

[Figure]

Figure S10 $RO_2$ fates under typical nighttime atmospheric conditions in the boreal forest in Finland. Conditions with both low (0.2 ppt, a) and high (1 ppt, b) $NO_3$ concentrations are considered.

Line 369 - 371: Please check the grammar in this sentence.

Response: We have rewritten this sentence as follows.

"As a result, a model simulation was conducted using a 10 times higher OH concentration ($5 \times 10^5$ molecules $cm^{-3}$). The concentration of $NO_3$ radicals was 1 ppt and the concentrations of other species were the same as the values mentioned above."

References

[revised manuscript text omitted]

---

## Author Comment (AC3)

We are grateful to the reviewer for the thoughtful comments on the manuscript. Our point-to-point responses to each comment are as follows (reviewer's comments are in black font and our responses are in blue font).

General Comments

This manuscript presents measurements of gas-phase organic peroxy radicals ($RO_2$), highly oxygenated organic molecules (HOMs), and dimeric compounds formed from oxidation of α-pinene by either $O_3$ or $NO_3 + O_3$ in a flow tube reactor made using a nitrate chemical ionization mass spectrometer ($NO_3$-CIMS), together with kinetic model simulations. The authors find that the formation of ultra-low and extremely low volatility organic compounds (ULVOC and ELVOC) measurable by $NO_3$-CIMS is significantly reduced in the $NO_3 + O_3$ system and further conclude that "the formation of new particles in the synergistic oxidation regime is substantially inhibited compared to the $O_3$-only regime." However, aerosol mass concentrations in the $NO_3 + O_3$ system were observed to be a factor-of-two higher than in the $O_3$-only system, directly contradicting this conclusion. Although the manuscript is well written, in many respects it replicates the work of Li et al. 2024 and Bates et al. 2022. For these reasons, I recommend that publication be considered only after the comments detailed below are addressed.

1. Table S1. Please specify how the initial α-pinene, cyclohexane, and $O_3$ concentrations were determined (i.e., measured, modeled, or estimated). Please add columns that report the modeled fractions of α-pinene that reacted with each oxidant (i.e., $O_3$, OH, and $NO_3$) as well as the modeled initial $NO_2$ concentrations.

Response: Thanks for the reviewer's comment. We have specified the determination methods of α-pinene, cyclohexane, and $O_3$ concentrations in the footnote of Table S1.

"The initial concentration of α-pinene was estimated according to its gas concentration in the canister and the dilution ratio in the flow tube, the concentration of cyclohexane was derived assuming that the cyclohexane in the gentle flow of ultra-high-purity $N_2$ bubbled through its liquid was saturated, and the $O_3$ concentration was measured with an ozone analyzer (T400, API)."

We have also added the modeled initial $NO_2$ concentration and the fractions of α-pinene that reacted with each oxidant in Table S1.

2. Figure 1. Figures 1a and 1b are redundant. Please replace Figure 1a with one that shows the signals of total $RO_2$, total monomer, and total dimer normalized by the total α-pinene reacted for both the $O_3 + NO_3$ and $O_3$-only systems, with the bars subdivided to indicate the fractions of CHO and CHON species. Please include a discussion of this figure (e.g., were normalized signals of total monomers and dimers higher in the $O_3 + NO_3$ or $O_3$-only system?) and revise L176–196 accordingly. Please also include a CIMS spectrum of an $O_3$-only experiment for comparison to Figure 1c.

Response: We have replaced Figure 1a with a new figure according to the reviewer's suggestion. In addition, we have provided a difference mass spectrum (i.e., mass spectrum in $O_3 + NO_3$ regime minus that in $O_3$-only regime) in Figure 1c, which highlights the changes in the species distribution in the synergistic oxidation regime compared to the $O_3$-only regime.

We have rewritten the discussion of this figure in Section 3.1 of the revised manuscript.

"The abundance of gas-phase $RO_2$ species and HOMs in different oxidation regimes is shown in Figure

1a. The species signals are normalized by the total reacted α-pinene in each regime. Compared to the $O_3$-only regime, the normalized signals of total $RO_2$ and HOMs decrease by 62 – 68% in the synergistic $O_3$ + $NO_3$ regime. Although $NO_3$ oxidation accounts for a considerable fraction of reacted α-pinene in the synergetic oxidation regime, the signal contributions of HOM-ONs are not significant. This might be due to the low sensitivity of nitrate-CIMS to the ONs formed involving $NO_3$ oxidation (Section 2.1). ……Figure 1c shows a difference mass spectrum highlighting the changes in species distribution between the two oxidation regimes. Almost all $C_xH_yO_z$-HOM species decrease significantly in the $O_3$ + $NO_3$ regime compared to the $O_3$-only regime. Besides, a large set of HOM-ON species are formed, despite their relatively low signals……."

[Figure]

Figure 1 Distributions of $RO_2$ and HOMs in the $O_3$-only and $O_3$ + $NO_3$ regimes. (a) Signals of total $RO_2$, as well as HOM monomers and dimers normalized by the reacted α-pinene in each oxidation regime (Exps 1-5, 7-11). (b) Relative changes in the normalized signals of $C_xH_yO_z$-HOMs in the $O_3$ + $NO_3$ regime versus the $O_3$-only regime. Ion signals are normalized to $\Delta[\alpha\text{-pinene}]_{O3}$ in each oxidation regime to highlight the suppression effect of the synergistic chemistry between $^{NO3}RO_2$ and $^{Cl}RO_2$ or $^{OH}RO_2$ on $C_xH_yO_z$-HOM formation. (c) Difference mass spectrum between the two oxidation regimes. The positive and negative peaks indicate the species with enhanced and decreased formation in the $O_3$ + $NO_3$ regime compared to the $O_3$-only regime, respectively.

3. CHON Dimers. Both Bates et al. 2022 and Li et al. 2024 observe significant (and often dominant)

contributions of $CHON_2$ dimers to total (CHO + CHON) dimer signals, yet in this work "HOM-ONs mainly consist of…$C_{20}$ dimers that only contain one nitrogen atom." Please include a discussion of potential explanations for these differences.

Response: In this study, some $CHON_2$ dimers were also observed in the $O_3 + NO_3$ regime, despite their much lower signals than CHON dimers. A potential explanation for the differences in the contribution of $CHON_2$ dimers to total dimer signals observed in different studies is the difference in the instrument sensitivity. In general, the nitrate-CIMS has lower sensitivities to ONs than to the $C_xH_yO_z$-HOM counterparts (Shen et al., 2022; Hyttinen et al., 2015). Bates et al. (2022) used $CF_3O^-$ as the reagent ion of CIMS. Its sensitivity to ONs might be significantly higher than the nitrate ion. Li et al. (2024) used CI-Orbitrap with ammonium or nitrate reagent ions to detect oxygenated organic molecules in the synergistic oxidation regime and found that both the signal intensity of ONs and their signal contribution to the total dimers were much larger when using ammonium as reagent ions. Particularly, the signal contribution of $CHON_2$ is significantly lower than CHON dimers. Despite both using nitrate regent ions, the nitrate CI-Orbitrap in Li et al. (2024) possibly exhibits higher sensitivities to ONs than the nitrate-CIMS in our study.

We have added a discussion regarding the instrument's sensitivity in Section 2.1.

"However, the highly oxygenated organic nitrates may have a significantly lower sensitivity compared to the $C_xH_yO_z$-HOM counterparts, given that the substitution of -OOH or -OH groups by $–ONO_2$ group in the molecule would reduce the number of H-bond donors, which is a key factor determining the sensitivity of the nitrate CIMS (Shen et al., 2022; Hyttinen et al., 2015). In addition, Li et al. (2024) used CI-Orbitrap with ammonium or nitrate reagent ions to detect oxygenated organic molecules in the synergistic $O_3 + NO_3$ regime, and found that both the signal intensity of ONs and their signal contribution to the total dimers were much larger when using ammonium as reagent ions."

We have also added a discussion about the reason why $CHON_2$ dimers were not significantly observed in this study in Section 3.1.

"The $CHON_2$ dimers were also observed in the $O_3 + NO_3$ regime, despite their much lower signals than CHON dimers, which is different from recent studies by Bates et al. (2022) and Li et al. (2024), who found $CHON_2$ dimers account for an important fraction of the total dimer signals in the synergistic oxidation regime. A potential explanation for this discrepancy is the difference in the instrument sensitivity in these studies (see Section 2.1). In general, the nitrate-CIMS has lower sensitivities to ONs than to the $C_xH_yO_z$-HOM counterparts (Shen et al., 2022; Hyttinen et al., 2015). Bates et al. (2022) used $CF_3O^-$ as the reagent ion of CIMS. Its sensitivity to ONs might be significantly higher than the nitrate reagent ion. In addition, Li et al. (2024) observed a significantly lower signal contribution of $CHON_2$ dimers using CI-Orbitrap with nitrate reagent ions than with ammonium ions. Despite both using nitrate regent ions, the nitrate CI-Orbitrap in Li et al. (2024) possibly exhibits higher sensitivities to ONs than the nitrate-CIMS in our study."

4. Trends in $O_3$- and OH-Derived $RO_2$. L226–229 report a larger decrease in the normalized signals of $C_{10}H_{15}O_x$-$RO_2$ than $C_{10}H_{17}O_x$-$RO_2$ in the $O_3 + NO_3$ vs. $O_3$-only system. Conversely, Li et al. 2024 report that "the measured $C_{10}H_{15}O_x$ rose with $NO_3$ radicals" while "$C_{10}H_{17}O_{5,7}$ radicals from OH chemistry decreased by a factor of 9." Please include a discussion of these discrepancies and potential explanations.

Response: Li et al. (2024) reported a slight increase in $C_{10}H_{15}O_x$-$RO_2$ with increasing $NO_3$ concentrations,

and indicated that this phenomenon was likely due to the additional $C_{10}H_{15}O_x$ production from the H-abstraction pathway of $NO_3$ oxidation in their experiments. However, during the $NO_3$ oxidation of monoterpenes, the rate constant for H-abstraction by $NO_3$ radicals is $(4 - 10) \times 10^{-17}$ cm$^3$ molecule$^{-1}$ s$^{-1}$, which is $10^3 - 10^4$ times lower than the rate constant for the $NO_3$ addition channel (Martinez et al., 1998). Besides, the subsequent reactions of $RO_2$ species formed from H-abstraction by $NO_3$ radicals should be very similar to those derived from H-abstraction by OH radicals, which was found not important for $C_xH_yO_z$-HOM formation in the absence of NO (Zang et al., 2023). Therefore, the H-abstraction of α-pinene by $NO_3$ radicals would have negligible influence on $C_{10}H_{15}O_x$ formation.

As Li et al. (2024) used a low α-pinene concentration and relatively high $O_3$ and $NO_3$ concentrations in their experiments, the secondary oxidation of aldehydes, such as the substantially formed pinonaldehyde, by $NO_3$ radicals might be important, which could contribute to the additional formation of $C_{10}H_{15}O_x$-$RO_2$. However, in the present study, the second-generation oxidation processes are strongly inhibited due to the excess of α-pinene, therefore the formation of secondary $C_{10}H_{15}O_x$-$RO_2$ is not important.

In addition, Li et al. (2024) reported that the fraction of α-pinene oxidized by OH radicals decreased from 44% in the $O_3$ oxidation system to 6% in the $O_3$+ $NO_3$ system mainly due to the depletion of OH radicals by $NO_2$ and the competitive consumption of α-pinene by $NO_3$ radicals, which resulted in a significant decrease in $C_{10}H_{17}O_{5,7}$ radicals from OH chemistry as observed in their experiments. However, in the present study, because of the excess of α-pinene, over 97% of OH radicals react with α-pinene and the depletion of OH by $NO_2$ is minor (0.2 – 1.3%) in the $O_3$ + $NO_3$ regime. The reduction in the reacted α-pinene by OH radicals is less than 10% compared to the $O_3$-only regime. As a result, a smaller decrease in $C_{10}H_{17}O_{5,7}$ radicals was observed in our study.

We have added the above discussions in Section 3.2 of the revised manuscript.

5. Figure 3. How/why were these particular $RO_2$ and HOM species selected? Why not report simulated ratios for all $RO_2$ and HOMs in Figure 2 as well as for total CI-$RO_2$, OH-$RO_2$, CI-HOM, and OH-HOM? Please reformat figure to make radicals open symbols and HOMs closed symbols.

Response: Thanks for the reviewer's comment. We have added more $RO_2$ and HOMs in Figure 3. However, as detailed oxidation mechanisms of α-pinene are still not well understood (especially for the $RO_2$ autoxidation), it is difficult to simulate all species well using one set of parameters. In addition, the autoxidation rate constants for the highly oxygenated $RO_2$ (with oxygen numbers larger than 11) are even more uncertain, thus we did not add them to the figure. Overall, the $RO_2$ and HOMs shown here have relatively high abundance. Although the data points seem more discrete with the addition of more compounds, it is still the case that the best model-measurement agreements are obtained when $k_{NO3+CI}$/ $k_{NO3+OH}$ is 10 – 100.

[Figure]

Figure 3. Measurement-model comparisons of the signal ratios of different $C_{10}$ $RO_2$ and HOMs in the synergistic $O_3 + NO_3$ regime vs. the $O_3$-only regime. The cross-reaction rate constant of $^{NO3}RO_2 + {^{Cl}RO_2}$ was set to $1 \times 10^{-12}$ cm$^3$ molecule$^{-1}$ s$^{-1}$ and the rate of $^{NO3}RO_2 + {^{OH}RO_2}$ was varied from $1 \times 10^{-11}$ cm$^3$ molecule$^{-1}$ s$^{-1}$ to $1 \times 10^{-14}$ cm$^3$ molecule$^{-1}$ s$^{-1}$ in the model.

6. $RO_2$ Rate Constants and Branching Ratios. This work sets the rate constant for $^{NO3}RO_2 + {^{Cl}RO_2}$ to $1 \times 10^{-12}$ cm$^3$ molec.$^{-1}$ s$^{-1}$ and then constrains the rate constant for $^{NO3}RO_2 + {^{OH}RO_2}$ to be $1 \times 10^{-13-14}$ cm$^3$ molec.$^{-1}$ s$^{-1}$. Bates et al. 2022 constrains the bulk rate constant for $^{NO3}RO_2$ self/cross reactions to be $1 \times 10^{-13}$ cm$^3$ molec.$^{-1}$ s$^{-1}$ with an upper limit of $1 \times 10^{-12}$ cm$^3$ molec.$^{-1}$ s$^{-1}$. Please include a discussion that justifies and compares the chosen rate constants. Additionally, Bates et al. 2022 report a branching fraction to the ROOR for $^{NO3}RO_2 + {^{NO3}RO_2}$ self/cross reactions of 16% while the ROOR branching fraction for the self-reaction of ethene-derived $RO_2$ was recently shown by Murphy et al. 2023 (DOI: 10.1039/D3EA00020F) to be over an order of magnitude higher than previously assumed (23% vs. 1%). What branching fraction to the ROOR was assumed for the kinetic modeling? Did it vary depending on the identity of the $RO_2$ (i.e., $^{NO3}RO_2$ vs. $^{OH}RO_2$ vs. $^{Cl}RO_2$)? Please include a sensitivity analysis that explores the impact of the assumed ROOR branching ratio(s) on the modeling results.

Response: Thanks for the reviewer's comments.

(1) Recently, Zhao et al. (2018) revealed the bulk rate constant for $^{Cl}RO_2$ and $^{OH}RO_2$ self/cross reactions to be $2 \times 10^{-12}$ cm$^3$ molecule$^{-1}$ s$^{-1}$, and Bates et al. (2022) constrained the rate constant for $^{NO3}RO_2$ self/cross reactions to be $1 \times 10^{-13} - 1 \times 10^{-12}$ cm$^3$ molecule$^{-1}$ s$^{-1}$. In the present study, a default rate constant of $2 \times 10^{-12}$ cm$^3$ molecule$^{-1}$ s$^{-1}$ was chosen for $^{NO3}RO_2 + {^{Cl}RO_2}$. Considering that there remains large uncertainty in this rate constant, we have conducted a sensitivity analysis to evaluate its influence on the ratio of $k_{NO3+Cl}/ k_{NO3+OH}$. It should be noted that the self/cross-reaction rate constants of $^{Cl}RO_2$ and $^{OH}RO_2$ are held constant at $2 \times 10^{-12}$ cm$^3$ molecule$^{-1}$ s$^{-1}$ (Zhao et al., 2018) in this analysis. As shown in Figure S6, when the $^{NO3}RO_2 + {^{Cl}RO_2}$ rate constant increase from $2 \times 10^{-13} - 2 \times 10^{-12}$ cm$^3$ molecule$^{-1}$ s$^{-1}$, the best agreements between modelled and measured signal ratios of $RO_2$ and HOMs are achieved consistently with a $k_{NO3+Cl}/ k_{NO3+OH}$ ratio of $10 - 100$. These results suggest that the uncertainty in the $^{NO3}RO_2 + RO_2$ kinetics would not alter the conclusion regarding the relative reaction efficiency of $^{NO3}RO_2 + {^{Cl}RO_2}$ versus $^{NO3}RO_2 + {^{OH}RO_2}$.

We have added the above discussion in Section S2 of the Supplement.

[Figure]

Figure S6 Measurement-model comparisons of the signal ratios of different $C_{10}$ $RO_2$ and HOMs in the synergistic $O_3$ + $NO_3$ regime vs. the $O_3$-only regime. The cross-reaction rate constant of $^{NO3}RO_2$ + $^{Cl}RO_2$ was set to $2 \times 10^{-13}$ cm$^3$ molecule$^{-1}$ s$^{-1}$ in (a), $1 \times 10^{-12}$ cm$^3$ molecule$^{-1}$ s$^{-1}$ in (b), $1.5 \times 10^{-12}$ cm$^3$ molecule$^{-1}$ s$^{-1}$ in (c), $2 \times 10^{-12}$ cm$^3$ molecule$^{-1}$ s$^{-1}$ in (d).

(2) Recent studies suggested that the ROOR dimer formation branching ratio from the highly oxygenated $RO_2$ are fast (Berndt et al., 2018; Molteni et al., 2019), therefore a relatively high dimer formation branching ratio of 50% was used in this study. This branching ratio does not change with different $RO_2$ cross reactions. To estimate the influence of dimer formation branching ratio on the simulated changes in $RO_2$ and related HOM concentrations in the synergistic $O_3$ + $NO_3$ regime vs. the $O_3$-only regime, we have conducted a sensitivity analysis of this ratio and added the following discussion to Section S3 of the Supplement.

"Currently, quantitative constraints on the ROOR dimer formation branching ratio are rather limited. Recent studies suggested that the dimer formation rates from the highly oxygenated $RO_2$ are fast (Berndt et al., 2018; Molteni et al., 2019), therefore a relatively high and consistent dimer formation branching ratio of 50% was used for different $RO_2$ (e.g., $^{Cl}RO_2$, $^{OH}RO_2$, $^{NO3}RO_2$) in this study. Considering the large uncertainties in this branching ratio, we conducted a sensitivity analysis to evaluate its influence on the relative changes in $RO_2$ and related HOM concentrations in the synergistic $O_3$ + $NO_3$ regime versus the $O_3$-only regime. As shown in Figure S8, as the dimer formation branching ratio increases from 9% to 50%, the variation in the abundance $C_xH_yO_z$-$RO_2$ and HOMs due to the concurrence of $NO_3$ oxidation changes slightly (< 9% and < 10%, respectively). These sensitivity analyses indicate that the uncertainties in the $RO_2$ autoxidation rate and dimer formation branching ratio slightly affect the simulated distribution of $RO_2$ and HOMs across different oxidation regimes but do not significantly change the $k_{NO3+Cl}/k_{NO3+OH}$ ratio obtained in this study."

In addition, we have added the following statement to Section 3.2 of the main text.

"Further sensitivity analyses on the rate constant and dimer formation branching ratio of $RO_2$ cross

reactions indicate that the uncertainties in these reaction kinetics do not alter the conclusion regarding the $k_{NO3+Cl}/k_{NO3+OH}$ ratio either (see details in Sections S2 and S3)."

[Figure]

Figure S8 Influences of the dimer formation branching ratio on the relative changes in $RO_2$ and related HOM concentrations in the synergistic $O_3 + NO_3$ regime vs. the $O_3$-only regime.

7. OH Scavenger Experiments. Based on results from the OH scavenger experiments, it is suggested that "the cross-reaction of $^{Cl}RO_2 + ^{NO3}RO_2$ is fast compared to that of $^{Cl}RO_2 + ^{Cl}RO_2$ and $^{Cl}RO_2 + ^{OH}RO_2$." However, the observed trends are determined by the relative reactivities (concentrations ´ rate constants) of the $^{NO3}RO_2$, $^{Cl}RO_2$, and $^{OH}RO_2$ toward reaction with $^{Cl}RO_2$. As such, without knowledge of the $RO_2$ concentrations, an assessment of the relative magnitudes of the rate constants cannot be made. That said, in order to observe both $C_{20}H_{30}O_x$ and $C_{20}H_{31}NO_x$ signals, the $^{Cl}RO_2 + ^{Cl}RO_2$ and $^{Cl}RO_2 + ^{NO3}RO_2$ reactions must competitive. As such, the qualitative statement in L292–294 is valid.

Response: We appreciate the reviewer's point. In the revised manuscript, we have changed the qualitative statement to "Such an enhanced production of $C_{20}H_{31}NO_x$ as compared to the slightly deceased formation of $C_{20}H_{30}O_x$ indicates that the $^{Cl}RO_2 + ^{NO3}RO_2$ reactions are competitive compared to the $^{Cl}RO_2 + ^{Cl}RO_2$ and $^{Cl}RO_2 + ^{OH}RO_2$ reactions."

8. Figure 4. Please report ratios for all $RO_2$, HOMs, and dimers in Figure 2. The vertical line in panel b is misplaced. The x-axis labels in the total column of panel c are mislabeled. The y-axis labels should be signals not concentrations. Please use the same color/labeling schemes in Figures 2 and 4.

Response: Thanks for the reviewer's suggestion. We have added all $RO_2$, HOMs, and HOM-ONs shown in Figures 1c and 2 to Figure 4. We have also corrected all the labelling and format issues pointed out by the reviewer.

[Figure]

Figure 4. Relative changes in signals of (a) $C_{10}$ RO$_2$, (b) $C_{10}$ HOMs, and (c) $C_{20}$ dimers due to the addition of 100 ppm cyclohexane as an OH scavenger in the synergistic $O_3$ + $NO_3$ regime (Exps 7 and 12).

9. Figure 5. Analogous to Figures 3b and 3c in Li et al. 2024, please include pie charts showing the fractional contributions of total (CHO + CHON) IVOC, SVOC, LVOC, ELVOC, and ULVOC to the total normalized signals measured in the $O_3$ + $NO_3$ and $O_3$-only systems. Please use the same color/labeling schemes in Figures 5 and 7.

Response: Thanks for the reviewer's comments. Because the nitrate-CIMS exhibits a relatively low sensitivity to the ONs in this study, the pie charts showing the fractional contributions of different species groups are not a very accurate representation of the volatility changes across the two oxidation regimes. As shown in figures below, although the contribution of ULVOCs decreases in the $O_3$ + $NO_3$ regime compared to the $O_3$-only regime, the contribution of ELVOCs increases in the $O_3$ + $NO_3$ regime. This phenomenon is due to the significant decrease in the highly abundant $C_{10}$ HOMs, resulting in a large reduction in LVOCs and SVOCs. Meanwhile, ONs contribute less to the LVOCs and SVOCs due to their low signals. As a result, the contribution of LVOCs decreases significantly, leading to a slightly increased contribution of ELVOCs in the $O_3$ + $NO_3$ regime. We noticed that in Li et al. (2024), the two pie charts showing the contributions of volatility classes reflected the pure $O_3$ and $NO_3$ chemistry, respectively, rather than the $O_3$-only and $O_3$ + $NO_3$ chemistry. Even so, the contributions of LVOCs, SVOCs, and IVOCs did not change significantly between the two systems in their study.

[Figure]

In the revised manuscript, we have used the same color and labelling schemes in Figures 5 and 7.

10. Compound Abundances. It is important to note that "abundances" (e.g., L311–313) are measured CIMS signals and that different compounds could potentially have different CIMS sensitivities.

Response: Thanks for the reviewer's comment.

In this study, we assume that the $C_xH_yO_z$-HOMs derived from ozonolysis and OH oxidation of α-pinene exhibit the same sensitivity in nitrate-CIMS. However, the highly oxygenated organic nitrates may have a significantly lower sensitivity compared to the $C_xH_yO_z$-HOM counterparts, given that the substitution of -OOH or -OH groups by –$ONO_2$ group in the molecule would reduce the number of H-bond donors, which is a key factor determining the sensitivity of the nitrate CIMS (Shen et al., 2022; Hyttinen et al., 2015).

Considering that different compounds could potentially have different CIMS sensitivities, we have conducted a sensitivity analysis by using different instrument sensitivities for different compounds to clarify their influences on the relative changes in $RO_2$ and HOMs in the $O_3$ + $NO_3$ regime versus the $O_3$-only regime. Taking a 10 times higher sensitivity to the compounds with an O/C ratio less than 0.7, the total signals are elevated in both oxidation regimes, but there remain significant decreases in total $RO_2$ and HOM signals in the synergistic oxidation regime compared to the $O_3$-only regime (Figure S4a). In addition, given that the sensitivity of nitrate-CIMS to ONs are relatively low, a 10 times higher sensitivity was also considered for the ONs. Under this condition, although ONs make a larger contribution to the total HOM monomers and dimers in the $O_3$ + $NO_3$ regime (Figure S4b), the signals of both total and $C_xH_yO_z$ $RO_2$ and HOMs still decrease significantly due to the presence of $NO_3$ oxidation. Therefore, different instrument sensitivities to $RO_2$ and HOMs with different oxygenation levels would not significantly influence the results (e.g., Figure 1) in this study.

We have added the above first paragraph to Section 3.1 of the main text and the second paragraph to Section S1 of the supplement.

We have also added the following statement to Section 3.1.

"Although there remain considerable uncertainties in instrument sensitivities to different compounds, sensitivity analyses suggest that varying the CIMS sensitivities to $RO_2$ and HOMs by a factor of 10 would not significantly influence their relative distribution across different oxidation regimes (see Section S1 for details)."

In addition, considering the high uncertainty in the instrument sensitivities to ONs, we have deleted the discussion regarding the abundance of ONs in L311-313.

[Figure]

Figure S4 Influences of different instrument sensitivities on the relative changes in $RO_2$ and HOMs in the synergistic oxidation regime versus the $O_3$-only regime. A 10 times higher instrument sensitivity to (a) compounds with O/C < 0.7 and (b) ONs was considered.

11. Figure 6. Please include measured particle-size distributions for the $O_3 + NO_3$ and $O_3$-only systems.

Response: We have added a figure showing the particle size distributions in different oxidation regimes as well as the relevant discussions to the revised manuscript.

"On the other hand, substantial formation of HOM-ONs is expected from the cross reactions of $^{NO3}RO_2$ with $^{Cl}RO_2$ and $^{OH}RO_2$ in the synergistic oxidation regime (Li et al., 2024; Bates et al., 2022), although their signals are relatively low due to the low sensitivity of nitrate-CIMS to ONs in this study. The newly formed HOM-ONs have relatively higher volatilities and are inefficient in initiating particle nucleation, but they are able to partition into the formed particles and contribute to the particle mass growth. Meanwhile, as the particle number concentration decreases drastically in the synergistic oxidation regime, more condensable vapors are available for each particle to grow to larger sizes (Figure 6b), which would in turn favor the condensation of more volatile organic species including ONs due to the reduced curvature effect of the larger particles, ultimately resulting in an increase in SOA mass concentrations."

[Figure]

Figure 6b. Size distributions of particles formed from the ozonolysis and synergistic $O_3 + NO_3$ oxidation of α-pinene (Exps 13-14).

12. New Particle Formation. Consistent with Li et al. 2024, this work finds that the presence of $NO_3$ radicals during α-pinene ozonolysis reduces the abundance of ELVOC and ULVOC measured in the gas

phase. However, in contrast to Li et al. 2024, this work observes a factor-of-two increase in aerosol mass concentrations in the $O_3 + NO_3$ vs. $O_3$-only system. Given that these experiments were conducted in the absence of seed aerosol, the higher aerosol mass loadings in the $O_3 + NO_3$ system indicate more efficient particle nucleation and growth, despite the reduced signals of gas-phase ELVOC and ULVOC measurable by $NO_3$-CIMS. The reduced particle number concentrations in the $O_3 + NO_3$ system are "ascribed to the suppressed formation of ULVOCs," however, enhanced coagulation seems more likely given the differences in mass loading. These results are also in contrast to Bates et al. 2022, which found that $O_3 + NO_3$ oxidation of α-pinene does not nucleate. However, they align with the seeded chamber experiments in Bates et al. 2022, which demonstrate that "high $^{NO3}RO_2 + RO_2$ contributions without any ozonolysis exhibited some of the highest measured SOA yields, suggesting perhaps that the $^{NO3}RO_2 + {}^{NO3}RO_2$ pathway on its own results in even higher SOA yields while $^{NO3}RO_2$ + other $RO_2$ pathways have lower yields." Please include a discussion of these discrepancies and potential explanations (e.g., efficient formation of ELVOCs and ULVOCs in $O_3 + NO_3$ system that are not measurable by $NO_3$-CIMS).

Response: Thanks for the reviewer's comment. This comment is partly addressed in our responses to last comment.

The presence of $NO_3$ radicals during α-pinene ozonolysis reduces the abundance of ULVOCs, which are the key species driving particle nucleation, thereby leading to a reduction in the particle number concentration in the $O_3 + NO_3$ regime. On the other hand, as discussed in our responses to last comment, substantial formation of HOM-ONs is expected from the cross reactions of $^{NO3}RO_2$ with $^{CI}RO_2$ and $^{OH}RO_2$ in the synergistic oxidation regime (Li et al., 2024; Bates et al., 2022), although their signals are relatively low due to the low sensitivity of nitrate-CIMS to ONs in this study. The newly formed HOM-ONs have relatively higher volatilities and are inefficient in initiating particle nucleation, but they are able to partition into the formed particles and contribute to the particle mass growth. Meanwhile, as the particle number concentration decreases drastically in the synergistic oxidation regime, more condensable vapors are available for each particle to grow to larger sizes (Figure 6b), which would in turn favor the condensation of more volatile organic species including ONs due to the reduced curvature effect of the larger particles, ultimately resulting in an increase in SOA mass concentrations. Recently, Bates et al. (2022) also found that in chamber experiments with seed particles, the SOA mass yields were significantly higher during α-pinene oxidation by $O_3 + NO_3$ than during ozonolysis, mainly due to the substantial formation and condensation of ON dimers. However, in the absence of seed particles, synergistic $O_3 + NO_3$ oxidation of α-pinene does not nucleate in their study. This phenomenon might be due to the high concentrations of $NO_2$ (72 ppb) and $O_3$ (102 ppb) as well as the relatively low concentration of α-pinene (27 ppb) in their experiments. As indicated by Bates et al. (2022), under this conditions $NO_3$ radicals were substantially formed and contributed to a dominant fraction (75%) of α-pinene oxidation, which strongly inhibited the production of low-volatility species and particle nucleation.

We have added the above discussions to the revised manuscript.

13. Termination Reactions. The term "termination reaction" is used throughout the manuscript to refer to $RO_2$ self/cross-reactions. In addition to radical termination to either alcohols and carbonyls or ROOR accretion products, however, alkoxy radical propagation is also possible. As such, please replace instances of "termination reaction" with $RO_2$ self/cross-reaction.

Response: Thanks for the reviewer's comment. We have replaced "termination reaction" with "self/crossreaction" or "cross-reaction" in the revised manuscript.

14. Atmospherically Relevant Simulations. Are the stated reductions in L351 compared to simulations with the same initial conditions but with $NO_3$ concentrations and formation rates set to zero? Is the same amount of α-pinene consumed in the simulations with and without $NO_3$? Please clarify. Please also compare with the atmospherically relevant modeling results in Bates et al. 2022.

Response: Thanks for the reviewer's comment. The reductions in $C_xH_yO_z$-HOMs in the $O_3$ + $NO_3$ regime are compared to the simulations with the same initial concentrations but with $NO_3$ concentrations and formation rates set to zero. The amount of α-pinene consumed by $O_3$ is the same with and without $NO_3$ oxidation, but the total consumption of α-pinene in the synergistic $O_3$ + $NO_3$ regime is larger than that in the $O_3$-only regime as a result of $NO_3$ oxidation.

We have added the following clarifications in the revised manuscript.

[revised manuscript text omitted]